# Last Iterate Convergence
# in Monotone Mean Field Games

**Noboru Isobe**
RIKEN AIP
Tokyo, Japan
noboru.isobe@riken.jp

**Kenshi Abe**
CyberAgent
Tokyo, Japan
abe_kenshi@cyberagent.co.jp

**Kaito Ariu**
CyberAgent
Tokyo, Japan
kaito_ariu@cyberagent.co.jp

## Abstract

In the Lasry–Lions framework, Mean-Field Games (MFGs) model interactions among an infinite number of agents. However, existing algorithms either require strict monotonicity or only guarantee the convergence of averaged iterates, as in Fictitious Play in continuous time. We address this gap with the following theoretical result. First, we prove that the last-iterated policy of a proximal-point (PP) update with KL regularization converges to an equilibrium of MFG under non-strict monotonicity. Second, we see that each PP update is equivalent to finding the equilibria of a KL-regularized MFG. We then prove that this equilibrium can be found using Mirror Descent (MD) with an exponential last-iterate convergence rate. Building on these insights, we propose the Approximate Proximal-Point (`APP`) algorithm, which approximately implements the PP update via a small number of MD steps. Numerical experiments on standard benchmarks confirm that the `APP` algorithm reliably converges to the unregularized mean-field equilibrium without time-averaging.

## 1 Introduction

Mean Field Games (MFGs) provide a simple and powerful framework for approximating the behavior of large populations of interacting agents. Formulated initially by Lasry and Lions (2007) and M. Huang et al. (2006), MFGs model the collective behavior of homogeneous agents in continuous time and state settings using partial differential equations (Cardaliaguet and Hadikhanloo 2017; Lavigne and Pfeiffer 2023; Inoue et al. 2023). The formulation of MFGs using Markov decision processes (MDPs) in Bertsekas and Shreve (1978) and Puterman (1994) has enabled the study of discrete-time and discrete-state models (Gomes et al. 2010).

In this context, a player's policy $\pi$, i.e., a probability distribution over actions, induces the so-called mean field $\mu$. This mean field $\mu$—namely, the distribution of all players' states—then affects both the state-transition dynamics and the rewards received by every agent. This simple formulation has broadened the applicability of MFGs to Multi-Agent Reinforcement Learning (MARL) (Yang et al. 2018; Guo et al. 2019; Angiuli et al. 2022; Zeman et al. 2023; Angiuli et al. 2024). Moreover, it has become possible to capture interactions among heterogeneous agents (Gao and Caines 2017; Caines and M. Huang 2019).

The applicability of MFGs to MARL drives research into the theoretical aspects of numerical algorithms for MFGs. Under fairly general assumptions, the problem of finding an equilibrium

39th Conference on Neural Information Processing Systems (NeurIPS 2025).

in MFGs is known to be PPAD-complete (Yardim et al. 2024). Consequently, it is essential to impose assumptions that allow for the existence of algorithms capable of efficiently computing an equilibrium. One such assumption is contractivity (Q. Xie et al. 2021; Anahtarci et al. 2023; Yardim et al. 2023). However, many MFG instances are known to be non-contractive in practice (Cui and Koeppl 2021). A more realistic assumption is the Lasry–Lions-type monotonicity employed in Pérolat et al. (2022), F. Zhang et al. (2023), and Yardim and He (2024), which intuitively implies that a player's reward monotonically decreases as more agents converge to a single state. Under the monotonicity assumption, Online Mirror Descent (OMD) has been proposed and widely adopted (Pérolat et al. 2022; Cui and Koeppl 2022; Laurière et al. 2022; Fabian et al. 2023). OMD, especially when combined with function approximation via deep learning, has enabled the application of MFGs to MARL (Yang and Wang 2020; K. Zhang et al. 2021; Cui et al. 2022).

Theoretically, *last-iterate convergence (LIC)* without time-averaging is particularly important in deep learning settings due to the constraints imposed by neural networks (NNs), as it ensures that the policy obtained in the last iteration converges. In NNs, computing the time-averaged policy as in the celebrated Fictitious Play method (Brown 1951; Perrin et al. 2020) may be less meaningful due to nonlinearity in the parameter space. This motivation has spurred significant research into developing algorithms that achieve LIC in finite $N$-player games, as seen in, e.g., Mertikopoulos et al. (2018), Piliouras et al. (2022), Abe et al. (2023), and Abe et al. (2024). However, in the case of MFGs, the results on LIC under realistic assumptions are limited. We refer the reader to read § 7 and Appendix A to review the existing results in detail.

We aim to develop a simple method to achieve LIC for MFGs with a realistic assumption. The first result of this paper is the development of a Proximal Point (PP) method using Kullback–Leibler (KL) divergence. We establish a novel convergence result in Theorem 3.1, showing that the PP method achieves LIC under the monotonicity assumption. When attempting to obtain convergence results in MFG, one faces the difficulty of controlling the mean field $\mu$, which changes along with the iterative updates of the policy $\pi$. We overcome this difficulty using the Łojasiewicz inequality, a classical tool from real analytic geometry.

We further propose the Approximate Proximal Point (APP) method to make the PP method feasible, which can be interpreted as an approximation of it. Here, we show that one iteration of the PP method corresponds to finding an equilibrium of the MFG regularized by KL divergence. This insight leads to the idea of approximating the iteration of PP by regularized Mirror Descent (RMD) introduced by F. Zhang et al. (2023). Our second theoretical result, presented in Theorem 4.3, is the LIC of RMD with an exponential rate. This result is a significant improvement over previous studies that only showed the convergence of the time-averaged policy or convergence at a polynomial rate. In the proof, the dependence of the mean field $\mu$ on the policy $\pi$ makes it difficult to readily exploit the Lipschitz continuity of the $Q$-function. We address this issue by utilizing the regularizing effect of the KL divergence.

Our experimental results also demonstrate LIC. The APP method can be implemented by making only a small modification to the RMD and experimentally converges to the (unregularized) equilibrium.

In summary, the contributions of this paper are as follows:

---

**Contributions**

(i) We present an algorithm based on the celebrated PP method and, for the first time, establish LIC for *non-strictly* monotone MFGs (Theorem 3.1).

(ii) We show that one iteration of the PP method is equivalent to solving the regularized MFG, which can be solved exponentially fast by RMD (Theorem 4.3).

(iii) Based on these two theoretical findings, we develop the APP method as an efficient approximation of the PP method (Algorithm 1).

---

The organization of this paper is as follows: In § 2, we review the fundamental concepts of MFGs. In § 3, we introduce the PP method and its convergence results. In § 4, we present the RMD algorithm and its convergence properties. Finally, in § 5, we propose a combined approximation method, demonstrating its convergence through experimental validation. § 7 provides a review of related works.

## 2 Problem setting and preliminary facts

**Notation:** For a positive integer $N \in \mathbb{N}$, $[N] := \{1, \ldots, N\}$. For a finite set $X$, $\Delta(X) := \{p \in \mathbb{R}_{\geq 0}^{|X|} \mid \sum_{x \in X} p(x) = 1\}$. For a function $f : X \to \mathbb{R}$ and a probability $\pi \in \Delta(X)$, $\langle f, \pi \rangle := \langle f(\bullet), \pi(\bullet) \rangle := \sum_{x \in X} f(x) \pi(x)$. For $p^0, p^1 \in \Delta(X)$, define the KL divergence $D_{\mathrm{KL}}(p^0, p^1) := \sum_{x \in X} p^0(x) \log \left( p^0(x)/p^1(x) \right)$, and the $\ell^1$ distance as $\| p^0 - p^1 \| := \sum_{x \in X} |p^0(x) - p^1(x)|$.

### 2.1 Mean-field games

Consider a model based *Mean-Field Game (MFG)* that is defined through a tuple $(\mathcal{S}, \mathcal{A}, H, P, r, \mu_1)$. Here, $\mathcal{S}$ is a finite discrete space of states, $\mathcal{A}$ is a finite discrete space of actions, $H \in \mathbb{N}_{\geq 2}$ is a time horizon, and $P = (P_h)_{h=1}^H$ is a sequence of transition kernels $P_h : \mathcal{S} \times \mathcal{A} \to \Delta(\mathcal{S})$, that is, if a player with state $s_h \in \mathcal{S}$ takes action $a_h \in \mathcal{A}$ at time $h \in [H]$, the next state $s_{h+1} \in \mathcal{S}$ will transition according to $s_{h+1} \sim P_h (\cdot \mid s_h, a_h)$. In addition, $r = (r_h)_{h=1}^H$ is a sequence of reward functions $r_h : \mathcal{S} \times \mathcal{A} \times \Delta(\mathcal{S}) \to [0, 1]$, and $\mu_1 \in \Delta(\mathcal{S})$ is an initial probability of state. Note that, in the context of theoretical analysis of the online learning method for MFG (Pérolat et al. 2022; F. Zhang et al. 2023), $P$ is assumed to be independent of the state distribution. It is reasonable to assume that at any time $h$, every state $s' \in \mathcal{S}$ is reachable:

**Assumption 2.1.** For each $(h, s') \in [H] \times \mathcal{S}$, there exists $(s, a) \in \mathcal{S} \times \mathcal{A}$ such that $P_h (s' \mid s, a) > 0$.

Note that it does *not* require that, for any state $s' \in \mathcal{S}$, it is reachable by *any* state-action pair $(s, a) \in \mathcal{S} \times \mathcal{A}$.

*Remark* 2.2. Our analysis excludes cases in which $P$ depends on $\mu$, as studied in Zeman et al. (2023) and Zeng et al. (2024)). These studies also rely on other conditions such as contraction or herding, which differ in nature from our monotonicity assumption. Extending the analysis to a $\mu$-dependent $P$ requires a different approach than that in the existing literature, e.g., Pérolat et al. (2022) and F. Zhang et al. (2023). A full treatment of the case is left for future work.

Given a policy $\pi$, the probabilities $m[\pi] = (m[\pi]_h)_{h=1}^H \in \Delta(\mathcal{S})^H$ of the state is recursively defined as follows: $m[\pi]_1 = \mu_1$ and

$$m[\pi]_h(s_h) = \sum_{s_{h-1} \in \mathcal{S}, a_{h-1} \in \mathcal{A}} \pi_{h-1} (a_{h-1} \mid s_{h-1}) P_{h-1} (s_h \mid s_{h-1}, a_{h-1}) m[\pi]_{h-1}(s_{h-1}), \quad (2.1)$$

if $h = 2, \ldots, H$. We aim to maximize the following cumulative reward

$$J(\mu, \pi) := \sum_{(h,s,a) \in [H] \times \mathcal{S} \times \mathcal{A}} \pi_h (a \mid s) \, m[\pi]_h(s) r_h(s, a, \mu_h), \quad (2.2)$$

with respect to the policy $\pi$, given a sequence of state distributions $\mu \in \Delta(\mathcal{S})^H$. The *mean-field equilibrium* defined below means the pair of probabilities $\mu$ and policies $\pi$ that achieves the maximum under the constraints (2.1).

**Definition 2.3.** A pair $(\mu^\star, \pi^\star) \in \Delta(\mathcal{S})^H \times (\Delta(\mathcal{A})^{\mathcal{S}})^H$ is a *mean-field equilibrium* if it satisfies (i) $J(\mu^\star, \pi^\star) = \max_{\pi \in \Delta(\mathcal{S})^H} J(\mu^\star, \pi)$, and (ii) $\mu^\star = m[\pi^\star]$. In addition, set $\Pi^\star \subset (\Delta(\mathcal{A})^{\mathcal{S}})^H$ as the set of all policies that are in mean-field equilibrium.

Under Assumptions 2.4 and 2.5 below, there exists a mean-field equilibrium, see the proof of Saldi et al. (2018, Theorem 3.3.) and Pérolat et al. (2022, Proposition 1.). Note that the equilibrium may not be unique if the inequality given below in Assumption 2.4 is non-strict. In other words, the set $\Pi^\star \subset (\Delta(\mathcal{A})^{\mathcal{S}})^H$ is not a singleton in general. As an illustrative example, one might consider the trivial case where $r \equiv 0$. Our goal is to construct an algorithm that approximates a policy in $\Pi^\star$.

In this paper, we focus on rewards $r$ that satisfy the following two typical conditions, which are also assumed in Perrin et al. (2020), Perrin et al. (2022), Pérolat et al. (2022), Fabian et al. (2023), and F. Zhang et al. (2023). The first one is *monotonicity* of the type introduced by Lasry and Lions (2007), which means, under a state distribution $\mu = (\mu_h)_{h=1}^H \in \Delta(\mathcal{S})^H$, if players choose a strategy—called a policy $\pi = (\pi_h)_{h=1}^H \in (\Delta(\mathcal{A})^{\mathcal{S}})^H$ to be planned—that concentrates on a state or action, they will receive a small reward.

**Assumption 2.4** (Weak monotonicity of $r$). For all $\pi, \widetilde{\pi} \in (\Delta(\mathcal{A})^{\mathcal{S}})^H$, it holds that

$$\sum_{h=1}^{H} \sum_{(s,a) \in \mathcal{S} \times \mathcal{A}} \left( r_h(s, a, \mu_h^{\pi}) - r_h(s, a, \mu_h^{\widetilde{\pi}}) \right) (\rho_h(s, a) - \widetilde{\rho}_h(s, a)) \leq 0, \qquad (2.3)$$

where we set $\mu^{\pi} = m[\pi]$, $\rho_h(s, a) := \pi_h(a \mid s) \mu_h^{\pi}(s)$ and $\widetilde{\rho}_h(s, a) := \widetilde{\pi}_h(a \mid s) \mu_h^{\widetilde{\pi}}(s)$.

A reward $r$ satisfying Assumption 2.4 is said to be *monotone*. Furthermore, $r$ is said to be *strictly monotone* if the equality in (2.3) holds only if $\pi = \widetilde{\pi}$. Although most of the previous papers provide theoretical analysis under strict monotonicity, this excludes the case where the transition is symmetric. We demonstrate that such structures inherently allow the existence of distinct policies generating identical state distributions, leading to the failure of strict monotonicity.

> ***Example (Failure of strict monotonicity in symmetric transitions).*** In general, *symmetry of* $P$ with respect to states in MFGs violates strict monotonicity, while preserving monotonicity. Consider an MFG with *symmetric transition dynamics*, e.g., consider an MDP defined on $\mathcal{S} = \{s_1, s_2\}$, $\mathcal{A} = \{a_1, a_2\}$, $H \geq 2$, $\mu_1 = \left( \frac{1}{2}, \frac{1}{2} \right)$. For each $h \in [H]$, the transition kernels are $P_h(s' \mid s, a = a_1) = \begin{pmatrix} 0.2 & 0.8 \\ 0.8 & 0.2 \end{pmatrix}$, $P_h(s' \mid s, a = a_2) = \begin{pmatrix} 0.7 & 0.3 \\ 0.3 & 0.7 \end{pmatrix}$. If we take the policy $\pi$ such that $\pi_h(a_1 \mid s) = 1$, $\pi_h(a_2 \mid s) = 0$, $\widetilde{\pi}_h(a_1 \mid s) = 0$, $\widetilde{\pi}_h(a_2 \mid s) = 1$, for all $s \in \mathcal{S}$ and $h \in [H]$, we can see that $m[\pi]_h = m[\widetilde{\pi}]_h = (0.5, 0.5)$ for all $h$. Let the reward be of the form $r_h(s, a, \mu) = R_h(s, a) - f(\mu(s))$ with a non-decreasing function $f: [0, 1] \to \mathbb{R}$ such as $f(x) = x$, which models a crowd that avoids overcrowding. Then the monotonicity condition holds for the case $\pi \neq \widetilde{\pi}$. However, strict monotonicity would demand that equality occur *only* if $\pi = \widetilde{\pi}$. In this example, whenever $m[\pi] = m[\widetilde{\pi}]$ (here the uniform distribution), the above sum is zero even if $\pi \neq \widetilde{\pi}$. Hence, the game is monotone but *not* strictly monotone. Such phenomena are limitations in games with balanced transitions. □

The second is the Lipschitz continuity of $r$ with respect to $\mu \in (\Delta(\mathcal{S}))^H$, which is standard in the field of MFGs (Cui and Koeppl 2021; Fabian et al. 2023; F. Zhang et al. 2023).

**Assumption 2.5** (Lipschitz continuity of $r$). There exists a constant $L$ such that for every $h \in [H]$, $s \in \mathcal{S}$, $a \in \mathcal{A}$, and $\mu, \mu' \in \Delta(\mathcal{S})$: $|r_h(s, a, \mu) - r_h(s, a, \mu')| \leq L\|\mu - \mu'\|$.

## 3 Proximal point-type method for MFG

This section presents an algorithm motivated by the Proximal Point (PP) method. Let $\lambda > 0$ be a sufficiently small positive number, roughly "the inverse of learning rate." In the algorithm proposed in this paper, we generate a sequence $\left( (\sigma^k, \mu^k) \right)_{k=0}^{\infty} \subset (\Delta(\mathcal{A})^{\mathcal{S}})^H \times \Delta(\mathcal{S})^H$ as

$$\sigma^{k+1} = \underset{\pi \in (\Delta(\mathcal{A})^{\mathcal{S}})^H}{\arg \max} \left\{ J(\mu^{k+1}, \pi) - \lambda D_{m[\pi]}(\pi, \sigma^k) \right\}, \quad \mu^{k+1} = m[\sigma^{k+1}], \qquad (3.1)$$

where $m$ is defined in (2.1) and $D_\mu(\pi, \sigma^k) := \sum_h \mathbb{E}_{s \sim \mu_h} \left[ D_{\mathrm{KL}}(\pi_h(s), \sigma_h^k(s)) \right]$ with a probability $\mu \in \Delta(\mathcal{S})^H$. If the initial policy $\pi^0$ has full support, i.e., $\min_{(h,s,a) \in [H] \times \mathcal{S} \times \mathcal{A}} \pi_h^0(a \mid s) > 0$, the rule (3.1) is well-defined, see Proposition C.1.

Interestingly, the rule (3.1) is similar to the traditional Proximal Point (PP) method with KL divergence in mathematical optimization and Optimal Transport, see Censor and Zenios (1992) and Y. Xie et al. (2019). Therefore, we also refer to this update rule as the PP method. The well-known (O)MD in Pérolat et al. (2022) can be viewed as a linearization of the objective $J$ inside (3.1). Consequently, PP— which uses the full, un-linearised $J$— is expected to be less sensitive to approximation error, resulting in more robust convergence under non-strict monotonicity than MD. On the other hand, unlike the traditional PP method, our method changes the objective function $J(\mu^k, \bullet): (\Delta(\mathcal{A})^{\mathcal{S}})^H \to \mathbb{R}$ with each iteration $k \in \mathbb{N}$. Therefore, it is difficult to derive a theoretical convergence result of our traditional method from traditional theory. See also Remark 3.3.

### 3.1 Last-iterate convergence result

The following theorem implies the last-iterate convergence of the policies generated by (3.1). Specifically, it shows that under the assumptions above, the sequence of policies converges to the equilibrium set. This result is crucial for the effectiveness of the algorithm in reaching an optimal policy.

**Theorem 3.1.** *Let $(\sigma^k)_{k=0}^{\infty}$ be the sequence defined by (3.1). In addition to Assumptions 2.1, 2.4, and 2.5, assume that the initial policy $\pi^0$ has full support, i.e., $\min_{(h,s,a)\in[H]\times\mathcal{S}\times\mathcal{A}}\pi_h^0(a\mid s) > 0$. Then, the sequence $(\sigma^k)_{k=0}^{\infty}$ converges to the set $\Pi^\star$ of equilibrium, i.e., $\lim_{k\to\infty}\mathrm{dist}(\sigma^k,\Pi^\star) = 0$, where we set $\mathrm{dist}(\sigma,\Pi^\star) := \inf_{\pi^\star\in\Pi^\star}\sum_{(h,s)\in[H]\times\mathcal{S}}\|\sigma_h(s) - \pi_h^\star(s)\|_1$ for $\sigma \in (\Delta(\mathcal{A})^{\mathcal{S}})^H$.*

Note that Theorem 3.1 no longer relies on the *strict*-monotonicity imposed in earlier works (Hadikhanloo and Silva 2019; Elie et al. 2020; Pérolat et al. 2022). Moreover, unlike the continuous-time results of Perrin et al. (2020) and Pérolat et al. (2022), it applies directly to the discrete-time scheme (3.1).

***Proof sketch of Theorem 3.1.*** If we accept the next lemma, we can easily prove Theorem 3.1:

**Lemma 3.2.** *Suppose Assumption 2.4. Then, for any equilibrium $(\mu^\star, \pi^\star)$ it holds that*
$$D_{\mu^\star}(\pi^\star, \sigma^{k+1}) - D_{\mu^\star}(\pi^\star, \sigma^k) \le J(\mu^\star, \sigma^{k+1}) - J(\mu^\star, \pi^\star) - D_{\mu^{k+1}}(\sigma^{k+1}, \sigma^k)$$
$$\le J(\mu^\star, \sigma^{k+1}) - J(\mu^\star, \pi^\star). \tag{3.2}$$

Lemma 3.2 implies that the KL divergence from an equilibrium point to the generated policy becomes smaller as the cumulative reward $J$ increases. We note that the function $J(\mu^\star, \bullet)\colon (\Delta(\mathcal{A})^{\mathcal{S}})^H \ni \pi \mapsto J(\mu^\star, \pi) \in \mathbb{R}$ is a polynomial, thus real-analytic. Then we apply (Łojasiewicz 1971, §18, Théorème 2) and find that there exist positive constants $\alpha$ and $C$ satisfying $J(\mu^\star, \pi) - J(\mu^\star, \pi^\star) \le -C(\mathrm{dist}(\pi, \Pi^\star))^\alpha$, for any $\pi \in (\Delta(\mathcal{A})^{\mathcal{S}})^H$. Combining the above two inequalities yields that $D_{\mu^\star}(\pi^\star, \sigma^{k+1}) - D_{\mu^\star}(\pi^\star, \sigma^k) \le -C(\mathrm{dist}(\sigma^{k+1}, \Pi^\star))^\alpha$. Thus, the telescoping sum of this inequality yields $\sum_{k=1}^{\infty}(\mathrm{dist}(\sigma^k, \Pi^\star))^\alpha \le \frac{D_{\mu^\star}(\pi^\star, \sigma^0)}{C} < +\infty$, which implies $\lim_{k\to\infty}\mathrm{dist}(\sigma^k, \Pi^\star) = 0$. □

*Remark* 3.3 (Challenges in the proof of Theorem 3.1). The technical difficulty in the proof lies in the term $D_{\mu^{k+1}}(\sigma^{k+1}, \sigma^k)$ in (3.2). If it were not dependent on $\mu$, that is, $D_{\mu^{k+1}} = D_{\mu^\star}$, then LIC would follow straightforwardly from $D_{\mu^\star}(\pi^\star, \sigma^{k+1}) - D_{\mu^\star}(\pi^\star, \sigma^k) \le -D_{\mu^\star}(\sigma^{k+1}, \sigma^k)$, where we use Definition 2.3 and the second line of (3.2). However, $D_{\mu^{k+1}}$ changes depending on $k$. Therefore, in the above proof, we have made a special effort to avoid using $D_{\mu^{k+1}}(\sigma^{k+1}, \sigma^k)$. One may have seen proofs employing the simple argument described above in games other than MFG, such as monotone games (Rosen 1965). The reason why such an argument is possible in monotone games is that the mean field $\mu$ does not appear. This difference makes it difficult to use the straightforward argument described above in MFGs.

## 4 Approximating proximal point with mirror descent in regularized MFG

As in the PP method, it is necessary to find $(\mu^{k+1}, \sigma^{k+1})$ at each iteration. However, it is difficult to exactly compute $(\mu^{k+1}, \sigma^{k+1})$ due to the implicit nature of (3.1). Therefore, this section introduces Regularized Mirror Descent (RMD), which approximates the solution $(\mu^{k+1}, \sigma^{k+1})$ for each policy $\sigma^k$. The novel result in this section is that the divergence between the sequence generated by RMD and the equilibrium decays exponentially as shown in Figure 1.

### 4.1 Approximation of the update rule of PP with regularized MFG

Interestingly, solving (3.1) corresponds to finding an equilibrium for *KL-regularized MFG* introduced in Cui and Koeppl (2021) and F. Zhang et al. (2023). We review the settings for the regularized MFG. For each parameter $\lambda > 0$ and policy $\sigma \in (\Delta(\mathcal{A})^{\mathcal{S}})^H$, which plays the role of $\sigma^k$ in (3.1), we define the *regularized cumulative reward* $J^{\lambda,\sigma}(\mu, \pi)$ for $(\mu, \pi) \in \Delta(\mathcal{S})^H \times (\Delta(\mathcal{A})^{\mathcal{S}})^H$ to be
$$J^{\lambda,\sigma}(\mu, \pi) := J(\mu, \pi) - \lambda D_{m[\pi]}(\pi, \sigma). \tag{4.1}$$
The assumption of full support is also imposed on $\sigma$:

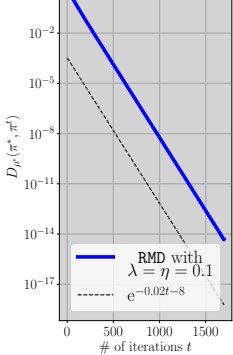

Figure 1: Behavior of RMD.

**Assumption 4.1.** The base $\sigma$ has full support, i.e., the minimum value given by

$$\sigma_{\min} := \min_{(h,s,a)\in[H]\times\mathcal{S}\times\mathcal{A}} \sigma_h(a \mid s)$$

is strictly positive.

For the reward $J^{\lambda,\sigma}$, we introduce a *regularized equilibrium*:

**Definition 4.2.** A pair $(\mu^*, \varpi^*) \in \Delta(\mathcal{S})^H \times (\Delta(\mathcal{A})^\mathcal{S})^H$ is *regularized equilibrium* of $J^{\lambda,\sigma}$ if it satisfies (i) $J^{\lambda,\sigma}(\mu^*, \varpi^*) = \max_{\pi\in\Delta(\mathcal{S})^H} J^{\lambda,\sigma}(\mu^*, \pi)$, and (ii) $\mu^* = m[\varpi^*]$.

Specifically, $(\mu^{k+1}, \sigma^{k+1})$ can be characterized as the regularized equilibrium of $J^{\lambda,\sigma^k}$ for $k \in \mathbb{N}$. Note that the equilibrium is unique under Assumption 4.1, see Appendix C.

In the next subsection, we will introduce RMD using *value functions*, which are defined as follows: for each $h \in [H]$, $s \in \mathcal{S}$, $a \in \mathcal{A}$, $\mu \in \Delta(\mathcal{S})^H$ and $\pi \in \Delta(\mathcal{A})^\mathcal{S}$, define the *state value function* $V_h^{\lambda,\sigma}: \mathcal{S} \times \Delta(\mathcal{S})^H \times (\Delta(\mathcal{A})^\mathcal{S})^H \to \mathbb{R}$ and the *state-action value function* $Q_h^{\lambda,\sigma}: \mathcal{S} \times \mathcal{A} \times \Delta(\mathcal{S})^H \times (\Delta(\mathcal{A})^\mathcal{S})^H \to \mathbb{R}$ as

$$V_h^{\lambda,\sigma}(s,\mu,\pi) := \mathbb{E}_{((s_l,a_l))_{l=h}^H}\left[\sum_{l=h}^H (r_l(s_l,a_l,\mu_l) - \lambda D_{\mathrm{KL}}(\pi_l(s_l),\sigma_l(s_l)))\right], \quad V_{H+1}^{\lambda,\sigma} \equiv 0, \quad (4.2)$$

$$Q_h^{\lambda,\sigma}(s,a,\mu,\pi) := r_h(s,a,\mu_h) + \mathbb{E}_{s_{h+1}\sim P(s,a,\mu_h)}[V_{h+1}^{\lambda,\sigma}(s_{h+1},\mu,\pi)]. \quad (4.3)$$

Here, the discrete-time stochastic process $((s_l,a_l))_{l=h}^H$ is induced recursively by $s_h = s$ and $s_{l+1} \sim P_l(s_l,a_l)$, $a_l \sim \pi_l(s_l)$ for each $l \in \{h,\ldots,H-1\}$ and $a_H \sim \pi_H(s_H)$. Note that the objective function $J^{\lambda,\sigma}$ in Definition 4.2 can be expressed as $J^{\lambda,\sigma}(\mu,\pi) = \mathbb{E}_{s\sim\mu_1}[V_1^{\lambda,\sigma}(s,\mu,\pi)]$.

## 4.2 An exponential convergence result

In this subsection, we introduce the iterative method for finding the regularized equilibrium proposed by F. Zhang et al. (2023) as RMD. The method constructs a sequence $((\pi^t,\mu^t))_{t=0}^\infty \subset (\Delta(\mathcal{A})^\mathcal{S})^H \times \Delta(\mathcal{S})^H$ approximating the regularized equilibrium of $J^{\lambda,\sigma}$ using the following rule:

$$\pi_h^{t+1}(s) = \arg\max_{p\in\Delta(\mathcal{A})}\left\{\frac{\eta}{1-\lambda\eta}\left(\left\langle Q_h^{\lambda,\sigma}(s,\bullet,\pi^t,\mu^t),p\right\rangle - \lambda D_{\mathrm{KL}}(p,\sigma_h(s))\right) - D_{\mathrm{KL}}(p,\pi_h^t(s))\right\},$$

$$\mu^{t+1} = m[\pi^{t+1}], \quad (4.4)$$

where $\eta > 0$ is another learning rate, and $Q_h^{\lambda,\sigma}$ is the state-action value function defined in (4.3). We give the pseudo-code of RMD in Algorithm 2. For the sequence of policies in RMD, we can establish the convergence result as follows:

> **Theorem 4.3.** *Let $((\mu^t,\pi^t))_{t=0}^\infty \subset \Delta(\mathcal{S})^H \times (\Delta(\mathcal{A})^\mathcal{S})^H$ be the sequence generated by (4.4), and $(\mu^*,\varpi^*) \in \Delta(\mathcal{S})^H \times (\Delta(\mathcal{A})^\mathcal{S})^H$ be the regularized equilibrium given in Definition 4.2. In addition to Assumptions 2.4, 2.5, and 4.1, suppose that $\eta \leq \eta^*$, where $\eta^* > 0$ is the upper bound of the learning rate defined in (D.5), which only depends on $\lambda$, $\sigma$, $H$ and $|\mathcal{A}|$. Then, the sequence $(\pi^t)_{t=0}^\infty$ satisfies that for $t \in \mathbb{N}$*
>
> $$D_{\mu^*}(\varpi^*,\pi^{t+1}) \leq \left(1 - \frac{\lambda\eta}{2}\right)D_{\mu^*}(\varpi^*,\pi^t),$$
>
> *which leads $D_{\mu^*}(\varpi^*,\pi^t) \leq D_{\mu^*}(\varpi^*,\pi^0)e^{-\lambda\eta t/2}$. Clearly, the inequality states that an approximate policy $\pi^t$ satisfying $D_{\mu^*}(\varpi^*,\pi^t) < \varepsilon$ can be obtained in $\mathcal{O}(\log(1/\varepsilon))$ iterations.*

*Remark* 4.4. While Theorem 4.3 provides an exponentially decreasing bound, the theoretical upper bound $\eta^*$ on the step size $\eta$ can be small, see (D.4) and (D.5) in detail.

This metric $D_{\mu^*}(\varpi^*,\pi^t)$ is widely used in F. Zhang et al. (2023) and Dong et al. (2025) because it provides an upper bound for the so-called exploitability $\mathrm{Exploit}(\pi) := \max_{\pi'} J(m[\pi],\pi') - J(m[\pi],\pi)$ as $\mathrm{Exploit}(\pi^t) = \mathcal{O}\left(\sqrt{D_{\mu^*}(\varpi^*,\pi^t)}\right)$ by the Lipschitz continuity of $V_1^{\lambda,\sigma}$. We also note that Theorem 4.3 improves upon the previous results by F. Zhang et al. (2023) and Dong et al. (2025) in the regime with a large number of iterations $t$. Indeed, the authors obtained $D_{\mu^*}(\varpi^*,\frac{1}{T}\sum_{t=1}^T \pi^t) \leq$

$\mathcal{O}\left(\lambda \log^2 T/\sqrt{T}\right)$ and $D_{\mu^*}(\varpi^*, \pi^{t+1}) \le {}^{H^3}/\lambda t$. On the other hand, these bounds for finite $t$ may be smaller since the constant $\eta$ inside our exponent could be small.

## 4.3 Intuition for exponential convergence: continuous-time version of RMD

The convergence of $(\pi^t)_{t=0}^\infty$ can be intuitively explained by considering a continuous limit $(\pi^t)_{t\ge0}$ with respect to the time $t$ of RMD. In this paragraph, we will use the idea of mirror flow (Krichene et al. 2015; Tzen et al. 2023; Deb et al. 2023) and continuous dynamics in games (Taylor and Jonker 1978; Mertikopoulos et al. 2018; Pérolat et al. 2021; Pérolat et al. 2022) to observe the exponential convergence of the flow to equilibrium. According to Deb et al. (2023, (2.1)), the continuous curve of $\pi$ should satisfy that

$$\frac{\mathrm{d}}{\mathrm{d}t}\pi_h^t(a \mid s) = \pi_h^t(a \mid s) \cdot \left(Q_h^{\lambda,\sigma}(s, a, \pi^t, \mu^t) - \lambda \log \frac{\pi_h^t(a \mid s)}{\sigma_h(a \mid s)}\right). \tag{4.5}$$

The flow induced by the dynamical system (4.5) converges to equilibrium *exponentially* as time $t$ goes to infinity.

> **Theorem 4.5.** *Let $\pi^t$ be a solution of (4.5) and $\varpi^*$ be a regularized equilibrium defined in Definition 4.2. Suppose that Assumption 2.4. Then*
> $$\frac{\mathrm{d}}{\mathrm{d}t}D_{\mu^*}(\varpi^*, \pi^t) \le -\lambda D_{\mu^*}(\varpi^*, \pi^t),$$
> *for all $t \ge 0$. Moreover, the inequality implies $D_{\mu^*}(\varpi^*, \pi^t) \le D_{\mu^*}(\varpi^*, \pi^0)\exp(-\lambda t)$.*

Technically, the non-Lipschitz continuity of the value function $Q_h^{\lambda,\sigma}(s, a, \bullet, \mu^t)$ in the right-hand side of (4.5) is non-trivial for the existence of the solution $\pi: [0, +\infty) \to (\Delta(\mathcal{A})^{\mathcal{S}})^H$ of the differential equation (4.5), see, e.g., Coddington and Levinson (1984). The proof of this existence and Theorem 4.5 are given in Appendix C.

## 4.4 Proof sketch of the convergence result for RMD

We return from continuous-time dynamics (4.5) to the discrete-time algorithm (4.4). The technical difficulty in the proof of Theorem 4.3 is the non-Lipschitz continuity of the value function $Q_h^{\lambda,\sigma}$ in (4.4), that is, the derivative of $Q_h^{\lambda,\sigma}(s, a, \pi, \mu)$ with respect to the policy $\pi$ can blow up as $\pi$ approaches the boundary of the space $(\Delta(\mathcal{A})^{\mathcal{S}})^H$ of probability simplices. We can overcome this difficulty as shown in the following sketch of proof:

> ***Proof sketch of Theorem 4.3.*** In a similar way to Theorem 4.5, we can obtain the following inequality with a discretization error:
>
> $$D_{\mu^*}(\varpi^*, \pi^{t+1}) - D_{\mu^*}(\varpi^*, \pi^t) \le -\lambda\eta D_{\mu^*}(\varpi^*, \pi^t) + \boxed{D_{\mu^*}(\pi^t, \pi^{t+1}),} \tag{4.6}$$
>
> where we use a property of KL divergence, see the proof in Appendix D. The remainder of the proof is almost entirely dedicated to showing that the above error term is sufficiently small and bounded compared to the other terms in (4.6). As a result, we obtain the following claim:
>
> > **Claim 4.6.** *Suppose that the learning rate $\eta$ is less than the upper bound $\eta^*$ in (D.5). Then*
> > $$\boxed{D_{\mu^*}(\pi^t, \pi^{t+1})} \le C\eta^2 D_{\mu^*}(\varpi^*, \pi^t),$$
> > *where $C > 0$ is the constant defined in (D.4), which satisfies $C\eta^* \le \lambda/2$.*
>
> The key to proving Claim 4.6 is leveraging another claim that, over the sequence $(\pi^t)_t$, the value function $Q_h^{\lambda,\sigma}$ behaves well, almost as if it were a Lipschitz continuous function, see Lemma D.3 for details. Therefore, applying Claim 4.6 to (4.6) completes the proof. $\qquad\square$

*Remark* 4.7 (Challenges in the proof of Theorem 4.3). The technical difficulty in the proof lies in the fact that the $Q$-function $Q_h^{\lambda,\sigma}(s, a, \pi^t, \mu^t)$ in the algorithm (4.4) depends on the mean field $\mu^t = m[\pi^t]$, which is determined forward by (2.1) from *past* times 1 to $h-1$. On the other hand,

| **Algorithm 1:** APP for MFG | **Algorithm 2:** RMD(MFG, $\pi^0, \lambda, \eta, \sigma^0, \tau$) |
|---|---|

**Algorithm 1:** APP for MFG

**Input:** MFG($\mathcal{S}, \mathcal{A}, H, P, r, \mu_1$), initial policy $\pi^0$, #iterations $N, \lambda > 0$

**1 Initialization:** Set $k \leftarrow 0, \sigma^k \leftarrow \pi^0$;

**2 while** $k < N$ **do**

**3**     Compute $(\mu^{k+1}, \sigma^{k+1})$ by solving

$$\begin{cases} \sigma^{k+1} = \text{RMD}(\text{MFG}, \sigma^k, \lambda, \eta, \sigma^k, \tau), \\ \mu^{k+1} = m[\sigma^{k+1}] \end{cases}$$

    Update $k \leftarrow k + 1$;

**Output:** $\sigma^k (\approx \pi^\star)$

**Algorithm 2:** RMD(MFG, $\pi^0, \lambda, \eta, \sigma^0, \tau$)

**1 Initialization:** Set $t \leftarrow 0, \pi^t \leftarrow \pi^0$, $\sigma \leftarrow \sigma^0$;

**2 while** $t < \tau$ **do**

**3**     Compute $\mu^t = m[\pi^t]$;

**4**     Compute $Q_h^{\lambda,\sigma}(s, a, \pi^t, \mu^t)$ by (4.3);

**5**     Compute $\pi^{t+1}$ as

$$\pi_h^{t+1}(a \mid s) \propto (\sigma_h(a \mid s))^{\lambda\eta}(\pi_h^t(a \mid s))^{1-\lambda\eta}$$
$$\cdot \exp\left(\eta Q_h^{\lambda,\sigma}(s, a, \pi^t, \mu^t)\right)$$

**6**     Update $t \leftarrow t + 1$;

**7 return** $\pi^t (\approx \varpi^*)$

the $Q$-function is also determined by the policy from *future* times $h + 1$ to $H$ through the dynamic programming principle given by (4.3). As a result, it becomes difficult to apply the backward induction argument, which is known in the context of MDPs and Markov games, to $Q$-functions. This difficulty is specific to MFGs and is not seen in other regularized games such as entropy-regularized zero-sum Markov games, where the $Q$-function depends only on future policies. Therefore, it is less feasible to directly apply the techniques of existing research, such as Cen et al. (2023), to RMD for MFGs. Our proof instead utilizes the properties of the KL divergence to deal with this difficulty.

### 4.5 APP: approximating PP updates with RMD

We recall that we need to develop an algorithm that efficiently approximates the update rule of the PP method since the rule (3.1) is intractable. To this end, we employ the *regularized* Mirror Descent (RMD) to solve the (*unregularized*) MFG as a substitute for the rule. Specifically, after repeating the RMD iteration (4.4) a sufficient number of times, we update the base distribution $\sigma$ using the most recently obtained policy $\sigma^{k+1}$. We call this method APP, which is summarized in Algorithm 1. In APP, updating the base seems like a small modification of RMD, but it is crucial for convergence. Without this update, we can only obtain regularized equilibria, which are generally different from our ultimate goal of unregularized equilibria. In fact, Definition 2.3, 4.2 and Assumption 2.4 yield that $J(\mu^\star, \pi^\star) - J(\mu^*, \varpi^*) \leq \lambda(D_{\mu^\star}(\pi^\star, \sigma) - D_{\mu^*}(\varpi^*, \sigma))$, which roughly implies that the gap between regularized and unregularized equilibria is $\mathcal{O}(\lambda)$. Experimental results in Cui and Koeppl (2021) also suggest that to find the (unregularized) equilibrium with a regularized algorithm, it is necessary to tune the hyperparameter $\lambda$ appropriately. Theoretically, the results we have established in Theorems 3.1 and 4.3 provide some convergence guarantees for APP. Empirically, the experimental results in the next section suggest that APP also achieves LIC. We conjecture that the rate of convergence for APP, as predicted by these experiments, may also be derived.

## 5 Numerical experiment

We numerically demonstrate that APP, which is the approximated version of (3.1), can achieve convergence to the mean-field equilibrium. We evaluate the convergence of APP using the Beach Bar Process introduced by Perrin et al. (2020), a standard benchmark for MFGs. In particular, the transition kernel $P$ in this benchmark gives a random walk on a one-dimensional discretized torus $\mathcal{S} = \{0, \ldots, |\mathcal{S}| - 1\}$, and the reward is set to be $r_h(s, a, \mu) = -|a|/|\mathcal{S}| - |s - |\mathcal{S}|/2|/|\mathcal{S}| - \log \mu_h(s)$ with $a \in \mathcal{A} := \{-1, \pm 0, +1\}$. Note that this benchmark satisfies the monotonicity assumption in Assumption 2.4. See Appendix F for further details. Figure 2 is a summary of the results of the experiment. The most notable aspect is the convergence of exploitability, as shown in Figure 2b. APP decreases the exploitability with each iteration when we update $\sigma$. Figure 2a and 2c illustrate the qualitative validity of the approximation achieved by APP. In this benchmark, the equilibrium is expected to lie at the vertices of the probability simplex. Therefore, RMD, which can shift the equilibrium to the interior of the probability simplex, seems unable to find the mean-field equilibrium

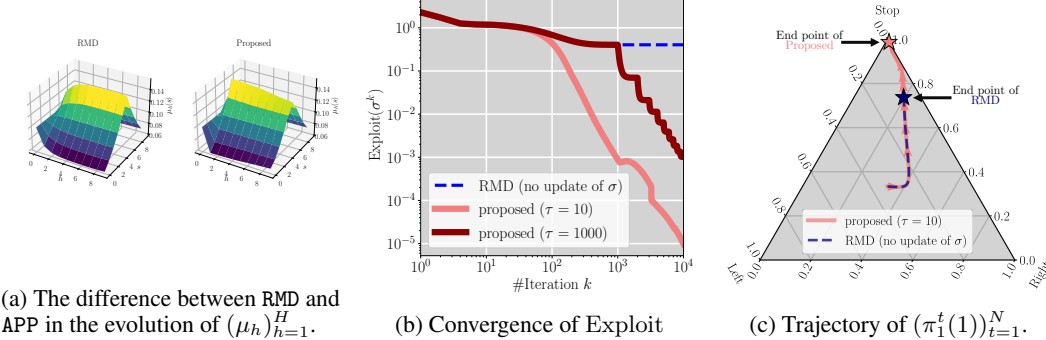

(a) The difference between RMD and APP in the evolution of $(\mu_h)_{h=1}^H$.

(b) Convergence of Exploit

(c) Trajectory of $(\pi_1^t(1))_{t=1}^N$.

Figure 2: Experimental results for Algorithm 1 for Beach Bar Process

accurately. On the other hand, the sequence $(\pi^t)_t$ of policies generated by APP shows a behavior that converges to the vertices.

## 6  Limitations

Our results provide the first asymptotic (Theorem 3.1) and exponential (Theorem 4.5) convergence guarantees for PP and RMD in non-strictly monotone (unregularized) MFGs under the model-based setting, assuming that the transition kernels and reward functions are available. This leaves open several important questions. First, we do not consider the more realistic scenario in which the transition of mean-field and reward must be learned from data, nor do we provide any sample-complexity or statistical guarantees, such as those in J. Huang et al. (2024), which would be required for rigorous model-free or data-driven applications. Second, our theoretical advantages are strictly in terms of iteration complexity under monotonicity: we establish faster convergence rates per iteration, but we do not claim any improvements in overall computational cost (for example, the cost of solving each PP subproblem or evaluating $Q$ in RMD), nor do we analyze how these methods scale with large state or action spaces in practice. Finally, although the proximal-point and mirror-descent structure of PP and RMD makes them, in principle, compatible with nonlinear function approximators, such as NNs, we have not studied approximation errors as in F. Zhang et al. (2023), stability issues, or empirical performance in high-dimensional or highly nonlinear settings.

Establishing LIC for APP remains open. We conjecture that a resolution will require proving a *uniform positive lower bound* on $\eta^*$ that guarantees LIC for RMD, which will improve the current estimate; see Remark 4.4.

By *synchronous feedback* we mean that, at (outer/inner) iterate $k$, all updates use the $Q$-function evaluated on the *current* pair $(\pi^k, \mu^k)$. In practice, feedback may be delayed or asynchronous. A natural adaptation is to evaluate $Q$ at a stale iterate, e.g., $Q(\pi^{\kappa^{(t)}}, \mu^{\kappa^{(t)}})$ at a delayed index $\kappa^{(t)}$. By analogy with asynchronous gradient play in zero-sum games (Ao et al. 2023), we expect last-iterate stability to persist under bounded staleness with a suitably reduced stepsize. A complete analysis in the MFG setting is left for future work.

## 7  Related works

As a result of the focus on the modeling potential of various population dynamics, there has been a significant increase in the literature on computations of equilibria in large-scale MFG, or so-called Learning in MFGs. We refer readers to read (Laurière et al. 2024) as a comprehensive survey of Learning in MFGs. Guo et al. (2019) and Anahtarci et al. (2020) developed a fixed-point iteration that alternately updates the mean-field $\mu$ and policy $\pi$, based on the algorithm of MDPs. They showed that this fixed-point iteration achieves LIC under a condition of contraction. However, it is known that the condition of contraction does not hold for many games in Cui and Koeppl (2021). In MFGs where the contraction assumption does not hold, it is observed that the fixed-point iteration oscillates in the case of linear-quadratic MFGs (Laurière 2021). Fictitious play, which averages mean fields or policies over time, was developed to prevent this oscillation. Hadikhanloo and Silva (2019), Elie et

al. (2020), and Perrin (2022) showed that the average in fictitious play converges to an equilibrium under the monotonicity assumption in Assumption 2.4. On the other hand, such time averaging has the disadvantage of slowing the experimental rate of convergence observed in Laurière et al. (2024) and making it difficult to scale up using deep learning.

Pérolat et al. (2022) applied Mirror Descent to MFG and developed a scalable method. This method has the practical benefits of being compatible with deep learning and is applicable to variants of variants (Laurière et al. 2022; Fabian et al. 2023). However, the theoretical guarantees are somewhat restrictive, as they often require strong assumptions like contraction for last-iterate convergence. In fact, they showed last-iterate convergence (LIC) of continuous-time algorithms under *strict* monotonicity assumptions. However, results for discrete-time settings or non-strict monotonicity are lacking. In addition to fictitious play and MD, methods using the actor-critic method (Zeng et al. 2024), value iteration (Anahtarci et al. 2020), multi-time scale (Angiuli et al. 2022; Angiuli et al. 2023; Angiuli et al. 2024) and semi-gradient method (C. Zhang et al. 2025) have been developed, but to the best of our knowledge, the theoretical convergence results of these methods require a condition of contraction. See the upper part of Table 1 for details.

Rather than focusing on the algorithm explained above, Cui and Koeppl (2021) focused on the problem setting of MFG and aimed to achieve a fast convergence of the algorithms by considering regularization of MFG. This type of regularization is typical in the case of MDPs and two-player zero-sum Markov games, where Mirror Descent achieves exponential convergence (Zhan et al. 2021; Cen et al. 2023). One expects similar convergence results for regularized MFGs, but the fast convergence results without strong assumptions have been limited so far. F. Zhang et al. (2023) and Dong et al. (2025) demonstrated polynomial convergence rates for MD under monotonicity. In addition, the authors in Q. Xie et al. (2021), Mao et al. (2022), Cui and Koeppl (2021), and Anahtarci et al. (2023) develop an algorithm that converges polynomially for regularized MFG, and they impose restrictive assumptions such as contraction and strict monotonicity. Appendix A provides an extensive review comparing existing results in Learning in MFGs.

# 8 Conclusion

This paper proposes the novel method to achieve LIC under the monotonicity (Assumption 2.4). The main idea behind the derivation of the method is to approximate the PP type method (3.1) using RMD. Theorem 3.1 implies that the PP method achieves LIC, and Theorem 4.3 establish the exponential convergence of RMD. A future task of this study is to prove the convergence rates of the combined method, APP.

## Acknowledgments and Disclosure of Funding

The first author was supported by JSPS KAKENHI Grant Numbers JP22KJ1002 and JP25H01453, as well as by JST ACT-X Grant Number JPMJAX25C2. Kaito Ariu was supported by JSPS KAKENHI Grant Numbers 23K19986 and 25K21291.

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

# A   Detailed Explanation of Related Works

Table 1: Summary of related work on convergence of iterative methods for MFGs

| | | Assumption | Discrete time | LIC |
|---|---|---|---|---|
| MFG | Guo et al. (2019) | Contract. | ✔ | - |
| | Elie et al. (2020), Hadikhanloo and Silva (2019) | Strict Mono. | ✔ | - |
| | Perrin et al. (2020) | Mono. | - | - |
| | Anahtarci et al. (2020) | Contract. | ✔ | ✔ |
| | Pérolat et al. (2022) | Strict Mono. | - | ✔ |
| | Geist et al. (2022) | Concavity | ✔ | ✔ |
| | Angiuli et al. (2022), Angiuli et al. (2023), Angiuli et al. (2024) | Contract. | ✔ | - |
| | Yardim et al. (2023) | Contract. | ✔ | ✔ |
| | Zeng et al. (2024) | Herding | ✔ | - |
| | C. Zhang et al. (2025) | Contract. | ✔ | ✔ |
| | Ours (Theorem 3.1) | Mono. | ✔ | ✔ |
| Regularized MFG | Q. Xie et al. (2021) | Contract. | ✔ | - |
| | Cui and Koeppl (2021) | Contract. | ✔ | ✔ |
| | Mao et al. (2022) | Contract. | ✔ | - |
| | Anahtarci et al. (2023) | Contract. | ✔ | ✔ |
| | F. Zhang et al. (2023) | Strict Mono. | ✔ | - |
| | Dong et al. (2025) | Mono. | ✔ | ✔ w/ poly. rate |
| | Ours (Theorem 4.5) | Mono. | ✔ | ✔ w/ exp. rate |

## A.1   Comparison with literature on MFGs

Based on Table 1, we will discuss the technical contributions made by this paper in Learning in MFGs below.

**Last-iterate convergence (LIC) results for MFGs:**   Pérolat et al. (2022) showed that Mirror Descent achieves LIC only under *strictly* monotone conditions, i.e., if the equality in the Lemma E.2 is satisfied only if $\pi = \widetilde{\pi}$. In contrast, our work establishes LIC even in *non-strictly* monotone scenarios. While the distinction regarding strictness might seem subtle, it is profoundly significant. Indeed, non-strictly monotone MFGs encompass the fundamental examples of finite-horizon Markov Decision Processes. Moreover, in strictly monotone cases, mean-field equilibria become unique. Consequently, as Zeng et al. (2024) also noted, strictly monotone rewards fail to represent MFGs with diverse equilibria.

**Regularized MFGs:**   Theorem 4.3, which supports the efficient execution of RMD, is novel in two respects: RMD achieves LIC, and the divergence to the equilibrium decays exponentially. Indeed, one of the few works that analyze the convergence rate of RMD states that the time-averaged policy $\frac{1}{T}\sum_{t=0}^{T}\pi^t$ up to time $T$ converges to the equilibrium in $\mathcal{O}(1/\varepsilon^2)$ iterations (F. Zhang et al. 2023). Additionally, although it is a different approach from MD, it is known that applying fixed-point iteration to regularized MFG achieves an exponential convergence rate under the assumption that the regularization parameter $\lambda$ is sufficiently large (Cui and Koeppl 2021). In contrast, our work includes the cases if $\lambda$ is small with $\eta < \eta^*$, where we note that $\eta^*$ depends on $\lambda$ though (D.5).

**Optimization-based methods for MFGs:** In addition to Mirror Descent and Fictitious Play, a new type of learning method using the characterization of MFGs as optimization problems has been proposed (Guo et al. 2024; Hu and J. Zhang 2024). In this work, the authors establish local convergence of the algorithms without the assumption of monotonicity. Specifically, it is proven that an optimization method can achieve LIC if the initial guess $\pi^0$ of the algorithm is sufficiently close to the Nash equilibrium, which cannot be verified a priori. In contrast, our convergence results state "global" convergence under the assumption of monotonicity of the reward. We note that the monotonicity can be checked before running the algorithms to ensure convergence of PP and RMD.

**Mean-field-aware methods for MFGs:** The authors in Zeng et al. (2024) and C. Zhang et al. (2025) have recently developed algorithms that sequentially update not only the policy $\pi$ but also the mean field $\mu$ and value function. These algorithms have advantages over conventional methods in terms of computational complexity. On the other hand, in theoretical analysis, restrictive assumptions such as contraction are still being used, and there is room for improvement under the monotonicity assumption.

### A.2 Comparison of MFG and related games

In research on the method of learning in games, regularization of games is often studied in order to improve extrapolation. For example, Geist et al. (2019) gave a unified convergence analysis method for regularized MDPs. (Leonardos et al. 2021) also discussed unique regularized equilibria of weighted zero-sum polymatrix games. On the other hand, it is a difficult task to apply the same theoretical analysis methods to MFG as to these games. In Remarks 3.3 and 4.7, we confirmed that the mean field $\mu$ in MFG can hinder convergence analysis. n the following two paragraphs, we will describe more specifically the difficulty of applying the methods used in other games to MFG.

**Sequential imperfect information game in Pérolat et al. (2021) vs. MFG:** Pérolat et al. (2021) focused on the reaching probability $\rho^\pi$ over histories in sequential imperfect information games, or extensive-form games. In contrast, we focused on the distribution of states $\mu = m[\pi]$ in MFGs. The dependency on $\pi$ is fundamentally different: $\rho$ depends on $\pi$ in a linear-like manner, while our $\mu$ has a highly nonlinear dependency on $\pi$ thorough the function $m$ defined in (2.1). Addressing this nonlinearity required novel techniques exploiting the inductive structure of (2.1) with respect to time $h$.

**MDP vs. MFG:** The known argument in Zhan et al. (2021, Lemma 6) cannot be directly applied to MFGs. The main reason is that the inner product $\langle Q^k(s), \pi^{k+1}(s) - p \rangle$ in the right-hand side of the three-point lemma concerns the policy at iteration index $k + 1$, not $k$. In our analysis (as shown on page 18), this term is transformed into $\langle Q^k(s), \pi^k(s) - p \rangle$, which allows us to apply a crucial lemma (Lemma E.4) that holds for MFGs. This transformation is non-trivial and essential for our analysis. In the three-point lemma, the term $D_{h_s}(\pi^{(k+1)}, \pi^{(k)})$ appears as a discretization error. In contrast, our analysis derives a reverse version $D_{\mu^*}(\pi^k, \pi^{k+1})$. This distinction is significant, especially for non-symmetric divergences such as the KL divergence. The reverse order in our analysis is crucial for the theoretical guarantees we provide.

## B Proof of Theorem 3.1

***Proof of Lemma 3.2.*** Let $(\mu^\star, \pi^\star)$ be a mean-field equilibrium defined in Definition 2.3. By the update rule (3.1) and Lemma E.1, we have

$$\left\langle Q_h^{\lambda,\sigma^k}(s, \bullet, \sigma^{k+1}, \mu^{k+1}) - \lambda \log \frac{\sigma_h^{k+1}(s)}{\sigma_h^k(s)}, (\pi_h^\star - \sigma_h^{k+1})(s) \right\rangle \leq 0,$$

for each $h \in [H]$, $s \in \mathcal{S}$ and $k \in \mathbb{N}$, i.e.,

$$D_{\mathrm{KL}}(\pi_h^\star(s), \sigma_h^{k+1}(s)) - D_{\mathrm{KL}}(\pi_h^\star(s), \sigma_h^k(s)) - D_{\mathrm{KL}}(\sigma_h^{k+1}(s), \sigma_h^k(s))$$
$$\leq \frac{1}{\lambda} \left\langle Q_h^{\lambda,\sigma^k}(s, \bullet, \sigma^{k+1}, \mu^{k+1}), (\sigma_h^{k+1} - \pi_h^\star)(s) \right\rangle. \tag{B.1}$$

Taking the expectation with respect to $s \sim \mu_h^\star$ and summing (B.1) over $h \in [H]$ yields

$$D_{\mu^\star}(\pi^\star, \sigma^{k+1}) - D_{\mu^\star}(\pi^\star, \sigma^k) + D_{\mu^\star}(\sigma^{k+1}, \sigma^k)$$

$$\leq \frac{1}{\lambda} \sum_{h=1}^{H} \mathbb{E}_{s \sim \mu_h^\star} \left[ \left\langle Q_h^{\lambda, \sigma^k}(s, \bullet, \sigma^{k+1}, \mu^{k+1}), (\sigma_h^{k+1} - \pi_h^\star)(s) \right\rangle \right].$$

By virtue of Lemmas E.2 and E.4, we further have

$$\sum_{h=1}^{H} \mathbb{E}_{s \sim \mu_h^\star} \left[ \left\langle Q_h^{\lambda, \sigma^k}(s, \bullet, \sigma^{k+1}, \mu^{k+1}), (\sigma_h^{k+1} - \pi_h^\star)(s) \right\rangle \right]$$

$$\leq J^{\lambda, \sigma^k}(\mu^{k+1}, \sigma^{k+1}) - J^{\lambda, \sigma^k}(\mu^{k+1}, \pi^\star) - \lambda D_{\mu^\star}(\pi^\star, \sigma^k) + \lambda D_{\mu^\star}(\sigma^{k+1}, \sigma^k)$$

$$\leq J^{\lambda, \sigma^k}(\mu^\star, \sigma^{k+1}) - J^{\lambda, \sigma^k}(\mu^\star, \pi^\star) - \lambda D_{\mu^\star}(\pi^\star, \sigma^k) + \lambda D_{\mu^\star}(\sigma^{k+1}, \sigma^k)$$

$$\leq J(\mu^\star, \sigma^{k+1}) - J(\mu^\star, \pi^\star) - \lambda D_{\mu^{k+1}}(\sigma^{k+1}, \sigma^k) + \lambda D_{\mu^\star}(\sigma^{k+1}, \sigma^k),$$

where we use the identity $J^{\lambda, \sigma^k}(\mu^\star, \pi) = J(\mu^\star, \pi) - \lambda D_{m[\pi]}(\pi, \sigma^k)$ for $\pi \in (\Delta(\mathcal{A})^{\mathcal{S}})^H$, and Definition 2.3. ∎

## C  Proof of Theorem 4.5

*Proof of Theorem 4.5.* Let $h^\star \colon \mathbb{R}^{|\mathcal{A}|} \to \mathbb{R}$ be the convex conjugate of $h$, i.e., $h^\star(y) = \sum_{a \in \mathcal{A}} \exp(y(a))$ for $y \in \mathbb{R}^{|\mathcal{A}|}$. From direct computations, we have

$$\frac{\mathrm{d}}{\mathrm{d}t} D_{\mu^*}(\varpi^*, \pi^t)$$

$$= \sum_{h=1}^{H} \mathbb{E}_{s \sim \mu_h^*} \left[ \frac{\mathrm{d}}{\mathrm{d}t} D_{\mathrm{KL}}(\varpi_h^*(s), \pi^t(s)) \right]$$

$$= \sum_{h=1}^{H} \mathbb{E}_{s \sim \mu_h^*} \left[ \left\langle 1 - \frac{\varpi_h^*(s)}{\pi_h^t(s)}, \frac{\mathrm{d}}{\mathrm{d}t} \pi_h^t(s) \right\rangle \right]$$

$$= \sum_{h=1}^{H} \mathbb{E}_{s \sim \mu_h^*} \left[ \left\langle 1 - \frac{\varpi_h^*(s)}{\pi_h^t(s)}, \pi_h^t(a \mid s) \left( Q_h^{\lambda, \sigma}(s, a, \pi^t, \mu^t) - \lambda \log \frac{\pi_h^t(a \mid s)}{\sigma_h(a \mid s)} \right) \right\rangle \right]$$

$$= \sum_{h=1}^{H} \mathbb{E}_{s \sim \mu_h^*} \left[ \left\langle (\pi_h^t - \varpi_h^*)(s), Q_h^{\lambda, \sigma}(s, \bullet, \pi^t, \mu^t) - \lambda \log \frac{\pi_h^t(a \mid s)}{\sigma_h(a \mid s)} \right\rangle \right]$$

$$= \sum_{h=1}^{H} \mathbb{E}_{s \sim \mu_h^*} \left[ \left\langle (\pi_h^t - \varpi_h^*)(s), Q_h^{\lambda, \sigma}(s, \bullet, \pi^t, \mu^t) \right\rangle \right] - \lambda \sum_{h=1}^{H} \mathbb{E}_{s \sim \mu_h^*} \left[ \left\langle (\pi_h^t - \varpi_h^*)(s), \log \frac{\pi_h^t(s)}{\sigma_h(s)} \right\rangle \right].$$

We apply Lemma E.4 for the first term and get

$$\sum_{h=1}^{H} \mathbb{E}_{s \sim \mu_h^*} \left[ \left\langle (\pi_h^t - \varpi_h^*)(s), Q_h^{\lambda, \sigma}(s, \bullet, \pi^t, \mu^t) \right\rangle \right] \tag{C.1}$$

$$= J^{\lambda, \sigma}(\mu^t, \pi^t) - J^{\lambda, \sigma}(\mu^t, \varpi^*) - \lambda D_{\mu^*}(\varpi^*, \sigma) + \lambda D_{\mu^*}(\pi^t, \sigma).$$

Similarly, we apply Lemma E.5 for the second term and get

$$\sum_{h=1}^{H} \mathbb{E}_{s \sim \mu_h^*} \left[ \left\langle (\pi_h^t - \varpi_h^*)(s), \log \frac{\pi_h^t(s)}{\sigma_h(s)} \right\rangle \right] = D_{\mu^*}(\pi^t, \sigma) - D_{\mu^*}(\varpi^*, \sigma) + D_{\mu^*}(\varpi^*, \pi^t). \tag{C.2}$$

Combining (C.1) and (C.2) yields

$$\frac{\mathrm{d}}{\mathrm{d}t} D_{\mu^*}(\varpi^*, \pi^t) = J^{\lambda, \sigma}(\mu^t, \pi^t) - J^{\lambda, \sigma}(\mu^t, \varpi^*) - \lambda D_{\mu^*}(\varpi^*, \pi^t).$$

By virtue of the definition of mean-field equilibrium and Lemma E.2, we find

$$J^{\lambda, \sigma}(\mu^t, \pi^t) - J^{\lambda, \sigma}(\mu^t, \varpi^*) \leq J^{\lambda, \sigma}(\mu^*, \pi^t) - J^{\lambda, \sigma}(\mu^*, \varpi^*) \leq 0.$$

Therefore, we obtain

$$\frac{\mathrm{d}}{\mathrm{d}t} D_{\mu^*}(\varpi^*, \pi^t) \leq -\lambda D_{\mu^*}(\varpi^*, \pi^t).$$

■

**Proposition C.1.** *Under Assumptions 2.1, 2.4, and 2.5, there exists a unique maximizer of* $J^{\lambda,\sigma^k}(\mu^k, \bullet)\colon (\Delta(\mathcal{A})^{\mathcal{S}})^H \to \mathbb{R}$ *for each* $k \in \mathbb{N}$.

Proposition C.1 also leads the uniqueness of the regularized equilibrium introduced in Definition 4.2. To elaborate further: Suppose there are two different regularized equilibria $(\mu_1^*, \varpi_1^*)$ and $(\mu_2^*, \varpi_2^*)$. If we assume $\varpi_1^* \neq \varpi_2^*$, the following contradiction arises: From Lemma E.2, we have

$$J^{\lambda,\sigma}(\mu_1^*, \varpi_1^*) + J^{\lambda,\sigma}(\mu_2^*, \varpi_2^*) \leq J^{\lambda,\sigma}(\mu_1^*, \varpi_2^*) + J^{\lambda,\sigma}(\mu_2^*, \varpi_1^*).$$

Additionally, from Proposition C.1, we know that $J^{\lambda,\sigma}(\mu_1^*, \varpi_1^*) \gneq J^{\lambda,\sigma}(\mu_1^*, \varpi_2^*)$ and $J^{\lambda,\sigma}(\mu_2^*, \varpi_2^*) \gneq J^{\lambda,\sigma}(\mu_2^*, \varpi_1^*)$. Adding these two inequalities gives us

$$J^{\lambda,\sigma}(\mu_1^*, \varpi_1^*) + J^{\lambda,\sigma}(\mu_2^*, \varpi_2^*) \gneq J^{\lambda,\sigma}(\mu_1^*, \varpi_2^*) + J^{\lambda,\sigma}(\mu_2^*, \varpi_1^*).$$

Therefore, $\varpi_1^* = \varpi_2^*$. Moreover, by the definition of regularized equilibria, $\mu_1^* = m[\varpi_1^*] = m[\varpi_2^*] = \mu_2^*$. This contradicts the assumption that the two equilibria are different. Thus, the equilibrium is unique.

The uniqueness of Proposition C.1 itself is a new result. The proof uses a continuous-time dynamics shown in Theorem 4.5, see Appendix C. In the following proof, we employ the same proof strategy as in Chill et al. (2010, Theorem 2.10). Before the proof, set $v_{s,h}^{\lambda,\sigma}(\pi) :=$ $\pi_h(a \mid s)\left(Q_h^{\lambda,\sigma}(s,a,\pi,m[\pi]) - \lambda \log \frac{\pi_h(a \mid s)}{\sigma_h(a \mid s)}\right)$ for $\pi \in (\Delta(\mathcal{A})^{\mathcal{S}})^H$.

***Proof of Proposition C.1.*** The existence is shown by a slightly modified version of (F. Zhang et al. 2023, Theorem 2). It remains to prove the uniqueness. Fix the regularized equilibrium $\varpi^* \in (\Delta(\mathcal{A})^{\mathcal{S}})^H$.

First of all, we prove the global existence of (4.5). By the local Lipschitz continuity of the right-hand side of the dynamics (4.5) and Picard–Lindelöf theorem, there exists a unique maximal solution $\pi$ of (4.5) with the initial condition $\pi|_{t=0} = \pi^0$. Namely, there exist $T \in (0, +\infty]$ and $\pi\colon [0,T) \to \mathbb{R}^{|\mathcal{A}|}$ such that $\pi$ is differentiable on $(0,T)$ and it holds that (4.5) for all $t \in (0,T)$. Thus, Theorem 4.5 ensures that

$$D_{\mu^*}(\varpi^*, \pi^t) + \lambda \int_0^t D_{\mu^*}(\varpi^*, \pi^\tau) \,\mathrm{d}\tau \leq D_{\mu^*}(\varpi^*, \pi^0) =: c < +\infty,$$

for every $t \in [0,T)$. As a result, the trajectory $\{\pi^t \in (\Delta(\mathcal{A})^{\mathcal{S}})^H \mid t \in [0,T)\}$ is included in $K_c := \{\pi \in (\Delta(\mathcal{A})^{\mathcal{S}})^H \mid D_{\mu^*}(\varpi^*, \pi) \leq c\}$. Note that $K_c$ is compact from Pinsker inequality.

Since the right-hand side of (4.5) is continuous on $K_c$, we obtain $\sup_{t \in [0,+\infty)} \left\| v_{s,h}^{\lambda,\sigma}(\pi^t) \right\| < +\infty$. Thus, the equation (4.5) implies $\left\| \frac{\mathrm{d}\pi^t}{\mathrm{d}t} \right\|$ is uniformly bounded on $[0,T)$. Hence, $\pi$ extends to a continuous function on $[0,T]$.

To obtain a contradiction, we assume $T < +\infty$. Then, there exists the solution $\pi'$ of (4.5) on a larger interval than $\pi$ with a new initial condition $\pi'|_{t'=T} = \pi^T$, which contradicts the maximality of the solution $\pi$.

Therefore, the limit $\lim_{t\to\infty} \pi^t$ exists and is equal to $\varpi^*$. Here, $\varpi^*$ is arbitrary, so the regularized equilibrium is unique. ■

## D   Proof of Theorem 4.3

We can easily show the following lemma by the optimality of $\pi^{t+1}$ in (4.4).

**Lemma D.1.** *It holds that*

$$\left\langle \eta\left(Q_h^{\lambda,\sigma}(s,\bullet,\pi^t,\mu^t) - \lambda \log \frac{\pi_h^{t+1}(s)}{\sigma_h(s)}\right) - (1-\lambda\eta)\log \frac{\pi_h^{t+1}(s)}{\pi_h^t(s)}, \delta \right\rangle = 0,$$

*for all* $\delta \in \mathbb{R}^{|\mathcal{A}|}$ *such that* $\sum_a \delta(a) = 0$.

We next show that $(\pi^t)_t$ is apart from the boundary of $\mathcal{A}$ as follows.

**Lemma D.2.** *Let $(\pi^t)_t$ be the sequence defined by* (4.4) *and $\varpi^*$ be the policy satisfies Definition 4.2. Assume that there exist vectors $w_h^\sigma$ and $w_h^0(s) \in \mathbb{R}^{|\mathcal{A}|}$ satisfying*

$$\lambda H \log \sigma_{\min} \leq w_h^\sigma\,(a \mid s) \leq -\lambda H \log \sigma_{\min}, \qquad \sigma_h\,(a \mid s) \propto \exp\left(\frac{w_h^\sigma\,(a \mid s)}{\lambda}\right),$$

$$2\lambda H \log \sigma_{\min} \leq w_h^0\,(a \mid s) \leq H, \qquad \pi_h^0\,(a \mid s) \propto \exp\left(\frac{w_h^0\,(a \mid s)}{\lambda}\right).$$

*for all $a \in \mathcal{A}.\pi^0 \in (\Delta(\mathcal{A})^{\mathcal{S}})^H$, $h \in [H]$ and $s \in \mathcal{S}$. Then, for any $h \in [H], s \in \mathcal{S}$, and $t \geq 0$, it holds that*

$$\max\left\{\left\|\log \pi_h^t(s)\right\|_\infty, \left\|\log \pi_h^*(s)\right\|_\infty\right\} \leq \frac{H(1 - \lambda \log \sigma_{\min})}{\lambda} + \log|\mathcal{A}|.$$

**Proof.** We first show that $\pi_h^t$ can be written as

$$\pi_h^t\,(a \mid s) \propto \exp\left(\frac{w_h^t\,(a \mid s)}{\lambda}\right), \tag{D.1}$$

for a vector $w_h^t(s) \in \mathbb{R}^{|\mathcal{A}|}$ satisfying $2\lambda H \log \sigma_{\min} \leq w_h^t\,(a \mid s) \leq H$. We prove it by induction on $t$. Suppose that there exist $t \in \mathbb{N}$ and $w_h^t$ satisfying (D.1). By the update rule (4.4), we have

$$\pi_h^{t+1}\,(a \mid s) \propto (\sigma_h\,(a \mid s))^{\lambda\eta}\big(\pi_h^t\,(a \mid s)\big)^{1-\lambda\eta} \exp\left(\eta Q_h^{\lambda,\sigma}(s,a,\pi^t,\mu^t)\right)$$

$$\propto \exp\left(\frac{\lambda\eta w_h^\sigma\,(a \mid s) + (1 - \eta\lambda)w_h^t\,(a \mid s) + \lambda\eta Q_h^{\lambda,\sigma}(s,a,\pi^t,\mu^t)}{\lambda}\right).$$

Set $w_h^{t+1}\,(a \mid s) := \lambda\eta w_h^\sigma\,(a \mid s) + (1 - \eta\lambda)w_h^t(a|s) + \lambda\eta Q_h^{\lambda,\sigma}(s,a,\pi^t,\mu^t)$, we get $\pi_h^{t+1}(a|s) \propto e^{\frac{w_h^{t+1}(a \mid s)}{\lambda}}$. From Lemma E.3 and the hypothesis of the induction, we get $2\lambda H \log \sigma_{\min} \leq w_h^{t+1}\,(a \mid s) \leq H$.

Then we have for any $a_1, a_2 \in \mathcal{A}$:

$$\frac{\pi_h^t\,(a_1 \mid s)}{\pi_h^t\,(a_2 \mid s)} = \exp\left(\frac{w_h^t\,(a_1 \mid s) - w_h^t\,(a_2 \mid s)}{\lambda}\right) \leq \exp\left(\frac{H(1 - \lambda \log \sigma_{\min})}{\lambda}\right).$$

It follows that:

$$\min_{a \in \mathcal{A}} \pi^t(a|s) \geq \exp\left(\frac{-H(1 - \lambda \log \sigma_{\min})}{\lambda}\right) \max_{a' \in \mathcal{A}} \pi_h^t\,(a \mid s) \geq |\mathcal{A}|^{-1} \exp\left(\frac{-H(1 - \lambda \log \sigma_{\min})}{\lambda}\right).$$

Therefore, we have:

$$\left\|\log \pi_h^t(s)\right\|_\infty \leq \frac{H(1 - \lambda \log \sigma_{\min})}{\lambda} + \log|\mathcal{A}|.$$

From Lemmas E.1 and E.3, we have for $\pi_h^*$ and $a_1, a_2 \in \mathcal{A}$:

$$\frac{\pi_h^*\,(a_1 \mid s)}{\pi_h^*\,(a_2 \mid s)} = \exp\left(\frac{Q_h^{\lambda,\sigma}(s,a_1,\pi^t,\mu^t) + w_h^\sigma\,(a_1 \mid s) - Q_h^{\lambda,\sigma}(s,a_2,\pi^t,\mu^t) - w_h^\sigma\,(a_2 \mid s)}{\lambda}\right)$$

$$\leq \exp\left(\frac{H(1 - \lambda \log \sigma_{\min})}{\lambda}\right),$$

and, we get $\left\|\log \pi_h^*(s)\right\|_\infty \leq \frac{H(1-\lambda \log \sigma_{\min})}{\lambda} + \log|\mathcal{A}|.$ ∎

**Lemma D.3.** *Let $G_h^{\lambda,\sigma}(s,a,\pi^t,\mu^t) := Q_h^{\lambda,\sigma}(s,a,\pi^t,\mu^t) - \lambda \log \frac{\pi_h^t\,(a \mid s)}{\sigma_h\,(a \mid s)}.$*

$$\left|G_h^{\lambda,\sigma}(s,a,\pi^t,\mu^t) - G_h^{\lambda,\sigma}(s,a',\pi^t,\mu^t)\right|$$

$$\leq 2L \sum_{l=h}^H \left\|\mu_l^t - \mu_l^*\right\|_1 + C^{\lambda,\sigma,H,|\mathcal{A}|}\big(E_h(a,\pi^t,\varpi^*) + E_h(a',\pi^t,\varpi^*)\big),$$

*for $a, a' \in \mathcal{A}$. Here,*

$$C^{\lambda,\sigma,H,|\mathcal{A}|} := 2\lambda|\mathcal{A}|e^{\frac{H(1-\lambda \log \sigma_{\min})}{\lambda}} + 2(1 + H) - \lambda(1 + 2H) \log \sigma_{\min} + 2\lambda \log |\mathcal{A}|,$$

*and*

$$E_h(a, \pi^t, \varpi^*) := \mathbb{E}\left[\sum_{l=h}^{H} \left\|\pi_l^*(s_l) - \pi_l^t(s_l)\right\|_1 \;\middle|\; \begin{array}{c} s_h = s, a_h = a, \\ s_{l+1} \sim P_l(s_l, a_l), \\ a_l \sim \varpi_l^*(s_l) \\ \text{for each } l \in \{h, \ldots, H\} \end{array}\right].$$

**Proof of Lemma D.3.** We first compute the absolute value as follows:

$$\left| G_h^{\lambda,\sigma}(s, a, \pi^t, \mu^t) - G_h^{\lambda,\sigma}(s, a', \pi^t, \mu^t) \right|$$

$$= \left| \left( Q_h^{\lambda,\sigma}(s, a, \pi^t, \mu^t) - \lambda \log \frac{\pi_h^t(a \mid s)}{\sigma_h(a \mid s)} \right) - \left( Q_h^{\lambda,\sigma}(s, a', \pi^t, \mu^t) - \lambda \log \frac{\pi_h^t(a' \mid s)}{\sigma_h(a' \mid s)} \right) \right|$$

$$\leq \left| \left( Q_h^{\lambda,\sigma}(s, a, \varpi^*, \mu^*) - \lambda \log \frac{\pi_h^t(a \mid s)}{\sigma_h(a \mid s)} \right) - \left( Q_h^{\lambda,\sigma}(s, a', \varpi^*, \mu^*) - \lambda \log \frac{\pi_h^t(a' \mid s)}{\sigma_h(a' \mid s)} \right) \right|$$

$$+ \left| \left( Q_h^{\lambda,\sigma}(s, a, \pi^t, \mu^t) - Q_h^{\lambda,\sigma}(s, a, \varpi^*, \mu^*) \right) - \left( Q_h^{\lambda,\sigma}(s, a', \pi^t, \mu^t) - Q_h^{\lambda,\sigma}(s, a', \varpi^*, \mu^*) \right) \right|.$$

(D.2)

By Lemmas D.2 and E.1, the first term of right-hand side in (D.3) can be computed as

$$\left| \left( Q_h^{\lambda,\sigma}(s, a, \varpi^*, \mu^*) - \lambda \log \frac{\pi_h^t(a \mid s)}{\sigma_h(a \mid s)} \right) - \left( Q_h^{\lambda,\sigma}(s, a', \varpi^*, \mu^*) - \lambda \log \frac{\pi_h^t(a' \mid s)}{\sigma_h(a' \mid s)} \right) \right|$$

$$= \left| \left( \lambda \log \frac{\varpi_h^*(a \mid s)}{\sigma_h(a \mid s)} - \lambda \log \frac{\pi_h^t(a \mid s)}{\sigma_h(a \mid s)} \right) - \left( \lambda \log \frac{\varpi_h^*(a' \mid s)}{\sigma_h(a' \mid s)} - \lambda \log \frac{\pi_h^t(a' \mid s)}{\sigma_h(a' \mid s)} \right) \right|$$

$$\leq \lambda \left( \left| \log \frac{\varpi_h^*(a \mid s)}{\pi_h^t(a \mid s)} \right| + \left| \log \frac{\varpi_h^*(a' \mid s)}{\pi_h^t(a' \mid s)} \right| \right)$$

$$\leq \lambda \left( \frac{1}{\varpi_{\min}^*} + \frac{1}{\min_{a \in \mathcal{A}} \pi_h^t(a \mid s)} \right) \left( \left| \varpi_h^*(a \mid s) - \pi_h^t(a \mid s) \right| + \left| \varpi_h^*(a' \mid s) - \pi_h^t(a' \mid s) \right| \right)$$

$$\leq 2\lambda |\mathcal{A}| \exp \left( \frac{H(1 - \lambda \log \sigma_{\min})}{\lambda} \right) \left( \left| \varpi_h^*(a \mid s) - \pi_h^t(a \mid s) \right| + \left| \varpi_h^*(a' \mid s) - \pi_h^t(a' \mid s) \right| \right).$$

(D.3)

By Proposition E.8 and Lemma E.6, the second term is bounded as

$$\left| \left( Q_h^{\lambda,\sigma}(s, a, \pi^t, \mu^t) - Q_h^{\lambda,\sigma}(s, a, \varpi^*, \mu^*) \right) - \left( Q_h^{\lambda,\sigma}(s, a', \pi^t, \mu^t) - Q_h^{\lambda,\sigma}(s, a', \varpi^*, \mu^*) \right) \right|$$

$$\leq 2L \sum_{l=h}^{H} \left\| \mu_l^t - \mu_l^* \right\|_1$$

$$+ C^{\lambda,\sigma}(\pi^t, \varpi^*) \, \mathbb{E}\left[\sum_{l=h+1}^{H} \left\|\pi_l^*(s_l) - \pi_l^t(s_l)\right\|_1 \;\middle|\; \begin{array}{c} s_{h+1} \sim P_h(\bullet \mid s, a), \\ s_{l+1} \sim P_l(s_l, a_l), \\ a_l \sim \varpi_l^*(s_l) \\ \text{for each } l \in \{h+1, \ldots, H\} \end{array}\right]$$

$$+ C^{\lambda,\sigma}(\pi^t, \varpi^*) \, \mathbb{E}\left[\sum_{l=h+1}^{H} \left\|\pi_l^*(s_l) - \pi_l^t(s_l)\right\|_1 \;\middle|\; \begin{array}{c} s_{h+1} \sim P_h(\bullet \mid s, a'), \\ s_{l+1} \sim P_l(s_l, a_l), \\ a_l \sim \varpi_l^*(s_l) \\ \text{for each } l \in \{h+1, \ldots, H\} \end{array}\right].$$

Furthermore, $C^{\lambda,\sigma}(\pi^t, \varpi^*)$ can be bounded as

$$C^{\lambda,\sigma}(\pi^t, \varpi^*) \leq 2 - \lambda \log \sigma_{\min} + 2\lambda \left( \frac{H(1 - \lambda \log \sigma_{\min})}{\lambda} + \log |\mathcal{A}| \right)$$

$$= 2(1 + H) - \lambda(1 + 2H) \log \sigma_{\min} + 2\lambda \log |\mathcal{A}|.$$

∎

**Proof of Theorem 4.3.** Set

$$C := 4H^2 \left( L^2 H^2 + \frac{\left( C^{\lambda,\sigma,H,|\mathcal{A}|} \right)^2}{|\mathcal{A}| \exp \left( \frac{H(1 - \lambda \log \sigma_{\min})}{\lambda} \right)} \right)$$

(D.4)

$$= 4H^2 \left( L^2 H^2 + \frac{\left( 2\lambda |\mathcal{A}| e^{\frac{H(1-\lambda \log \sigma_{\min})}{\lambda}} + 2(1+H) - \lambda(1+2H)\log \sigma_{\min} + 2\lambda \log |\mathcal{A}| \right)^2}{|\mathcal{A}| e^{\frac{H(1-\lambda \log \sigma_{\min})}{\lambda}}} \right)$$

$$\eta^* = \min \left\{ \frac{1}{2H\left(L + C^{\lambda,\sigma,H,|\mathcal{A}|}\right)}, \frac{\lambda}{2C} \right\}, \tag{D.5}$$

where $C^{\lambda,\sigma,H,|\mathcal{A}|}$ is the constant defined in Lemma D.3. We prove the inequality by induction on $t$.

**(I) Base step $t = 0$:**  It is obvious.

**(II) Inductive step:**  Suppose that there exists $t \in \mathbb{N}$ such that $\pi^t \in \Omega$. Lemma D.1 yields that

$$D_{\mu^*}(\varpi^*, \pi^{t+1}) - D_{\mu^*}(\varpi^*, \pi^t) - D_{\mu^*}(\pi^t, \pi^{t+1})$$

$$= \sum_{h=1}^{H} \mathbb{E}_{s \sim \mu_h^*} \left[ \left\langle \log \frac{\pi_h^t(s)}{\pi_h^{t+1}(s)}, (\varpi_h^* - \pi_h^t)(s) \right\rangle \right]$$

$$= - \sum_{h=1}^{H} \mathbb{E}_{s \sim \mu_h^*} \left[ \left\langle \frac{\eta}{1 - \lambda \eta} \left( Q_h^{\lambda,\sigma}(s, \bullet, \pi^t, \mu^t) - \lambda \log \frac{\pi_h^{t+1}(s)}{\sigma_h(s)} \right), (\varpi_h^* - \pi_h^t)(s) \right\rangle \right]$$

$$= - \frac{\eta}{1 - \lambda \eta} \underbrace{\sum_{h=1}^{H} \mathbb{E}_{s \sim \mu_h^*} \left[ \left\langle Q_h^{\lambda,\sigma}(s, \bullet, \pi^t, \mu^t), (\varpi_h^* - \pi_h^t)(s) \right\rangle \right]}_{=:\mathrm{I}} \tag{D.6}$$

$$+ \frac{\lambda \eta}{1 - \lambda \eta} \sum_{h=1}^{H} \mathbb{E}_{s \sim \mu_h^*} \left[ \left\langle \log \frac{\pi_h^{t+1}(s)}{\sigma_h(s)}, (\varpi_h^* - \pi_h^{t+1})(s) \right\rangle \right]$$

$$\leq - \frac{\eta}{1 - \lambda \eta} \left( \lambda D_{\mu^*}(\varpi^*, \sigma) - \lambda D_{\mu^*}(\pi^{t+1}, \sigma) \right)$$

$$+ \frac{\lambda \eta}{1 - \lambda \eta} \left( D_{\mu^*}(\varpi^*, \sigma) - D_{\mu^*}(\varpi^*, \pi^{t+1}) - D_{\mu^*}(\pi^{t+1}, \sigma) \right)$$

$$\leq - \frac{\lambda \eta}{1 - \lambda \eta} D_{\mu^*}(\varpi^*, \pi^{t+1}),$$

where I is bounded from below as follows: By Lemma E.4, we get

$$\mathrm{I} = J^{\lambda,\sigma}(\mu^{t+1}, \varpi^*) - J^{\lambda,\sigma}(\mu^{t+1}, \pi^{t+1}) + \lambda D_{\mu^*}(\varpi^*, \sigma) - \lambda D_{\mu^*}(\pi^{t+1}, \sigma).$$

By virtue of the definition of mean-field equilibrium and Lemma E.2, we find

$$J^{\lambda,\sigma}(\mu^{t+1}, \varpi^*) - J^{\lambda,\sigma}(\mu^{t+1}, \pi^{t+1}) \geq J^{\lambda,\sigma}(\mu^*, \varpi^*) - J^{\lambda,\sigma}(\mu^*, \pi^{t+1}) \geq 0.$$

Then, we obtain

$$\mathrm{I} \geq \lambda D_{\mu^*}(\varpi^*, \sigma) - \lambda D_{\mu^*}(\pi^{t+1}, \sigma).$$

For the last term $D_{\mu^*}(\pi^t, \pi^{t+1})$ of the leftmost hand of (D.6), we can employ a similar argument to (Abe et al. 2023, Lemma 5.4), that is, we can estimate $D_{\mu^*}(\pi^t, \pi^{t+1})$ as follows: Set $G(a) := G_h^{\lambda,\sigma}(s, a, \pi^t, \mu^t) = Q_h^{\lambda,\sigma}(s, a, \pi^t, \mu^t) - \lambda \log \frac{\pi_h^t(a \mid s)}{\sigma_h(a \mid s)}$. Note that $\max_{a,a' \in \mathcal{A}} |G(a') - G(a)| \leq$

$\eta^{*-1}$ by Lemma D.3. By the update rule (4.4) and concavity of the logarithmic function log, we have

$$D_{\mu^*}(\pi^t, \pi^{t+1})$$

$$= \sum_{h=1}^{H} \mathbb{E}_{s \sim \mu_h^*}\left[\sum_{a \in \mathcal{A}} \pi_h^t(a \mid s) \log \frac{\pi_h^t(a \mid s)}{\pi_h^{t+1}(a \mid s)}\right]$$

$$= \sum_{h=1}^{H} \mathbb{E}_{s \sim \mu_h^*}\left[\sum_{a \in \mathcal{A}} \pi_h^t(a \mid s) \log \frac{\sum_{a' \in \mathcal{A}} (\sigma_h(a' \mid s))^{\lambda \eta}(\pi_h^t(a' \mid s))^{1-\lambda \eta} \exp\left(\eta Q_h^{\lambda,\sigma}(s, a', \pi^t, \mu^t)\right)}{(\sigma_h(a \mid s))^{\lambda \eta}(\pi_h^t(a \mid s))^{-\lambda \eta} \exp\left(\eta Q_h^{\lambda,\sigma}(s, a, \pi^t, \mu^t)\right)}\right]$$

$$= \sum_{h=1}^{H} \mathbb{E}_{s \sim \mu_h^*}\left[\sum_{a \in \mathcal{A}} \pi_h^t(a \mid s) \log \frac{\sum_{a' \in \mathcal{A}} \pi_h^t(a' \mid s) \exp\left(\eta Q_h^{\lambda,\sigma}(s, a', \pi^t, \mu^t) - \lambda \eta \log \frac{\pi_h^t(a' \mid s)}{\sigma_h(a' \mid s)}\right)}{\exp\left(\eta Q_h^{\lambda,\sigma}(s, a, \pi^t, \mu^t) - \lambda \eta \log \frac{\pi_h^t(a \mid s)}{\sigma_h(a \mid s)}\right)}\right]$$

$$\leq \sum_{h=1}^{H} \mathbb{E}_{s \sim \mu_h^*}\left[\log \sum_{a \in \mathcal{A}} \pi_h^t(a \mid s) \frac{\sum_{a' \in \mathcal{A}} \pi_h^t(a' \mid s) \exp\left(\eta Q_h^{\lambda,\sigma}(s, a', \pi^t, \mu^t) - \lambda \eta \log \frac{\pi_h^t(a' \mid s)}{\sigma_h(a' \mid s)}\right)}{\exp\left(\eta Q_h^{\lambda,\sigma}(s, a, \pi^t, \mu^t) - \lambda \eta \log \frac{\pi_h^t(a \mid s)}{\sigma_h(a \mid s)}\right)}\right].$$

$$\tag{D.7}$$

If we take $\eta$ to be $\eta \leq \eta^*$, it follows that
$$\eta(G(a') - G(a)) \leq 1,$$
for $a, a' \in \mathcal{A}$. Thus, we can use the inequality $e^x \leq 1 + x + x^2$ for $x \leq 1$ and obtain

$$D_{\mu^*}(\pi^t, \pi^{t+1})$$

$$\leq \sum_{h=1}^{H} \mathbb{E}_{s \sim \mu_h^*}\left[\log \sum_{a,a' \in \mathcal{A}} \pi_h^t(a \mid s) \pi_h^t(a' \mid s) e^{\eta(G(a') - G(a))}\right]$$

$$\leq \sum_{h=1}^{H} \mathbb{E}_{s \sim \mu_h^*}\left[\log \sum_{a,a' \in \mathcal{A}} \pi_h^t(a \mid s) \pi_h^t(a' \mid s) \left(1 + \eta(G(a') - G(a)) + \eta^2(G(a') - G(a))^2\right)\right]$$

$$= \sum_{h=1}^{H} \mathbb{E}_{s \sim \mu_h^*}\left[\log \sum_{a,a' \in \mathcal{A}} \pi_h^t(a \mid s) \pi_h^t(a' \mid s) \left(1 + (G(a') - G(a))^2\right)\right]$$

$$= \sum_{h=1}^{H} \mathbb{E}_{s \sim \mu_h^*}\left[\log \left(1 + \eta^2 \sum_{a,a' \in \mathcal{A}} \pi_h^t(a \mid s) \pi_h^t(a' \mid s) (G(a') - G(a))^2\right)\right]$$

$$\leq \eta^2 \sum_{h=1}^{H} \mathbb{E}_{s \sim \mu_h^*}\left[\sum_{a,a' \in \mathcal{A}} \pi_h^t(a \mid s) \pi_h^t(a' \mid s) (G(a') - G(a))^2\right].$$

By Lemma D.3, we can see that

$$\sum_{a,a' \in \mathcal{A}} \pi_h^t(a \mid s) \pi_h^t(a' \mid s) (G(a') - G(a))^2$$

$$\leq \sum_{a,a' \in \mathcal{A}} \pi_h^t(a \mid s) \pi_h^t(a' \mid s) \left(2L \sum_{l=h}^{H} \|\mu_l^t - \mu_l^*\|_1 + C^{\lambda,\sigma,H,|\mathcal{A}|}\left(E_h(a, \pi^t, \varpi^*) + E_h(a', \pi^t, \varpi^*)\right)\right)^2$$

$$\leq \sum_{a,a' \in \mathcal{A}} \pi_h^t(a \mid s) \pi_h^t(a' \mid s) \left(8L^2 \left(\sum_{l=h}^{H} \|\mu_l^t - \mu_l^*\|_1\right)^2 + 4\left(C^{\lambda,\sigma,H,|\mathcal{A}|}\right)^2 \left(E_h^2(a, \pi^t, \varpi^*) + E_h^2(a', \pi^t, \varpi^*)\right)\right)$$

$$\leq 8L^2 H \sum_{l=h}^{H} \|\mu_l^t - \mu_l^*\|_1^2 + 8\left(C^{\lambda,\sigma,H,|\mathcal{A}|}\right)^2 \sum_{a \in \mathcal{A}} \pi_h^t(a \mid s) E_h^2(a, \pi^t, \varpi^*)$$

$$= 8L^2 H \sum_{l=h}^{H} \left\| \mu_l^t - \mu_l^* \right\|_1^2 + 8 \left( C^{\lambda,\sigma,H,|\mathcal{A}|} \right)^2 \sum_{a \in \mathcal{A}} \frac{\pi_h^t(a \mid s)}{\varpi_h^*(a \mid s)} \varpi_h^*(a \mid s) E_h^2(a, \pi^t, \varpi^*)$$

$$\leq 8L^2 H \sum_{l=h}^{H} \left\| \mu_l^t - \mu_l^* \right\|_1^2 + \frac{8 \left( C^{\lambda,\sigma,H,|\mathcal{A}|} \right)^2}{|\mathcal{A}| \exp \left( \frac{H(1 - \lambda \log \sigma_{\min})}{\lambda} \right)} \sum_{a \in \mathcal{A}} \varpi_h^*(a \mid s) E_h^2(a, \pi^t, \varpi^*)$$

$$\leq 8L^2 H \sum_{l=h}^{H} \left\| \mu_l^t - \mu_l^* \right\|_1^2 + \frac{8H \left( C^{\lambda,\sigma,H,|\mathcal{A}|} \right)^2}{|\mathcal{A}| \exp \left( \frac{H(1 - \lambda \log \sigma_{\min})}{\lambda} \right)} \sum_{l=h}^{H} \mathbb{E}_{s_l \sim \mu_l^*} \left[ \left\| \pi_l^*(s_l) - \pi_l^t(s_l) \right\|_1^2 \right]$$

$$\leq 8L^2 H \sum_{l=h}^{H} \left\| \mu_l^t - \mu_l^* \right\|_1^2 + \frac{4H \left( C^{\lambda,\sigma,H,|\mathcal{A}|} \right)^2}{|\mathcal{A}| \exp \left( \frac{H(1 - \lambda \log \sigma_{\min})}{\lambda} \right)} D_{\mu^*}(\varpi^*, \pi^t).$$

Moreover, [Lemma E.6](#) bounds $\sum_{l=h}^{H} \left\| \mu_l^t - \mu_l^* \right\|_1^2$ as

$$\sum_{l=h}^{H} \left\| \mu_l^t - \mu_l^* \right\|_1^2 \leq H \sum_{l=h}^{H} \sum_{k=0}^{l-1} \mathbb{E}_{s_k \sim \mu_k^*} \left[ \left\| \pi_k^*(s_k) - \pi_k^t(s_k) \right\|^2 \right] \leq \frac{1}{2} H^2 D_{\mu^*}(\varpi^*, \pi^t).$$

Therefore, we finally obtain

$$D_{\mu^*}(\varpi^*, \pi^{t+1}) \leq \left( 1 - \lambda \eta + C \eta^2 \right) D_{\mu^*}(\varpi^*, \pi^t) \leq \left( 1 - \frac{1}{2} \lambda \eta \right) D_{\mu^*}(\varpi^*, \pi^t), \qquad \text{(D.8)}$$

where we use $C\eta \leq C\eta^* \leq 1/2$. ∎

# E  Useful Lemmas

For Mean-field games, one can write down the *Bellman optimality equation* as follows: for a function $Q' : \mathcal{S} \to \Delta(\mathcal{A})$, a policy $\pi' : \mathcal{S} \to \Delta(\mathcal{A})$, $\sigma' : \mathcal{S} \to \Delta(\mathcal{A})$ and $s \in \mathcal{S}$ set

$$f_s^{\sigma'}(Q', \pi') = \langle Q'(s), \pi'(s) \rangle - \lambda D_{\mathrm{KL}}(\pi'(s), \sigma'(s)). \qquad \text{(E.1)}$$

**Lemma E.1.** *Let $(\mu^*, \varpi^*)$ be equilibrium in the sense of [Definition 4.2](#). Then, it holds that*

$$\varpi_h^*(s) = \underset{p \in \Delta(\mathcal{A})}{\arg \max} \, f_s^{\sigma_h} \left( Q_h^{\lambda,\sigma}(s, \bullet, \varpi^*, \mu^*), p \right) \propto \sigma_h(\bullet \mid s) \exp \left( \frac{Q_h^{\lambda,\sigma}(s, \bullet, \varpi^*, \mu^*)}{\lambda} \right),$$

*for each $s \in \mathcal{S}$ and $h \in [H]$. Moreover,*

$$\left\langle Q_h^{\lambda,\sigma}(s, \bullet, \varpi^*, \mu^*) - \lambda \log \frac{\pi_h^*(s)}{\sigma_h(s)}, \delta \right\rangle = 0,$$

*for all $\delta \in \mathbb{R}^{|\mathcal{A}|}$ such that $\sum_a \delta(a) = 0$.*

**Proof.** See the Bellman optimality equation (e.g., (Agarwal et al. [2022](#), Theorem 1.9)). ∎

**Lemma E.2.** *Under [Assumption 2.4](#), it holds that, for all $\pi, \widetilde{\pi} \in (\Delta(\mathcal{A})^{\mathcal{S}})^H$,*
$$J^{\lambda,\sigma}(m[\pi], \pi) + J^{\lambda,\sigma}(m[\widetilde{\pi}], \widetilde{\pi}) - J^{\lambda,\sigma}(m[\pi], \widetilde{\pi}) - J^{\lambda,\sigma}(m[\widetilde{\pi}], \pi) \leq 0,$$
*where $m$ is defined in ([2.1](#)).*

**Proof of [Lemma E.2](#).** The proof is similar to (F. Zhang et al. [2023](#), §H). Set $\mu = m[\pi]$ and $\widetilde{\mu} = m[\widetilde{\pi}]$. One can obtain that

$$J^{\lambda,\sigma}(m[\pi], \pi) + J^{\lambda,\sigma}(m[\widetilde{\pi}], \widetilde{\pi}) - J^{\lambda,\sigma}(m[\pi], \widetilde{\pi}) - J^{\lambda,\sigma}(m[\widetilde{\pi}], \pi)$$

$$= (J^{\lambda,\sigma}(\mu, \pi) - J^{\lambda,\sigma}(\widetilde{\mu}, \pi)) + (J^{\lambda,\sigma}(\widetilde{\mu}, \widetilde{\pi}) - J^{\lambda,\sigma}(\mu, \widetilde{\pi}))$$

$$= \sum_{h=1}^{H} \sum_{s_h \in \mathcal{S}} m[\pi]_h(s_h) \sum_{a_h \in \mathcal{A}} \pi_h(a_h \mid s_h) (r_h(s_h, a_h, \mu_h) - r_h(s_h, a_h, \widetilde{\mu}_h))$$

$$+ \sum_{h=1}^{H} \sum_{s_h \in \mathcal{S}} m[\widetilde{\pi}]_h(s_h) \sum_{a_h \in \mathcal{A}} \widetilde{\pi}_h(a_h \mid s_h) (r_h(s_h, a_h, \widetilde{\mu}_h) - r_h(s_h, a_h, \mu_h))$$

$$= \sum_{h,s,a} (\pi_h(a \mid s)\,\mu_h(s) - \widetilde{\pi}_h(a \mid s)\,\widetilde{\mu}_h(s))(r_h(s_h, a_h, \mu_h) - r_h(s_h, a_h, \widetilde{\mu}_h)),$$

and the right-hand side of the above inequality is less than 0 by Assumption 2.4. ∎

**Lemma E.3.** *Let $V_h^{\lambda,\sigma}$ be the state value function defined in (4.2) and $Q_h^{\lambda,\sigma}$ be the state action value function defined in (4.3). For any $s \in \mathcal{A}$, $a \in \mathcal{A}$, and $h \in [H]$, it holds that*

$$\lambda(H - h + 1)\log \sigma_{\min} \leq V_h^{\lambda,\sigma}(s, \mu, \pi) \leq H - h + 1,$$
$$\lambda(H - h + 1)\log \sigma_{\min} \leq Q_h^{\lambda,\sigma}(s, a, \mu, \pi) \leq H - h + 2.$$

*Proof.* We prove the inequalities by backward induction on $h$. By definition, we have

$$V_h^{\lambda,\sigma}(s, \mu, \pi)$$
$$= \mathbb{E}\left[\sum_{l=h}^{H} (r_l(s_l, a_l, \mu_l) - \lambda D_{\mathrm{KL}}(\pi_l(s_l), \sigma_l(s_l))) \,\middle|\, s_h = s\right]$$
$$= \langle r_h(s, \bullet, \mu_h), \pi_h(s)\rangle - \lambda D_{\mathrm{KL}}(\pi_h(s_h), \sigma_h(s_h))$$
$$\quad + \sum_{s_{h+1} \in \mathcal{S}} V_{h+1}^{\lambda,\sigma}(s_{h+1}, \mu, \pi) \sum_{a_h \in \mathcal{A}} P_h(s_{h+1} \mid s, a_h)\,\pi_h(a_h \mid s)$$
$$\leq 1 + \max_{s_{h+1} \in \mathcal{S}} V_{h+1}^{\lambda,\sigma}(s_{h+1}, \mu, \pi),$$

and

$$V_h^{\lambda,\sigma}(s, \mu, \pi)$$
$$= \langle r_h(s, \bullet, \mu_h), \pi_h(s)\rangle - \lambda D_{\mathrm{KL}}(\pi_h(s_h), \sigma_h(s_h))$$
$$\quad + \sum_{s_{h+1} \in \mathcal{S}} V_{h+1}^{\lambda,\sigma}(s_{h+1}, \mu, \pi) \sum_{a_h \in \mathcal{A}} P_h(s_{h+1} \mid s, a_h)\,\pi_h(a_h \mid s)$$
$$\geq \lambda \log \sigma_{\min} + \max_{s_{h+1} \in \mathcal{S}} V_{h+1}^{\lambda,\sigma}(s_{h+1}, \mu, \pi).$$

Then, we have

$$V_h^{\lambda,\sigma}(s, \mu, \pi) \in [\lambda(H - h + 1)\log \sigma_{\min}, H - h + 1],$$

by the induction. The definition of $Q_h^{\lambda,\sigma}$ in (4.3) immediately yields the bound. ∎

**Lemma E.4.** *For all $\pi, \widetilde{\pi} \in (\Delta(\mathcal{A})^{\mathcal{S}})^H$, it holds that*

$$\sum_{h=1}^{H} \mathbb{E}_{s \sim m[\widetilde{\pi}]_h}\left[\left\langle (\pi_h - \widetilde{\pi}_h)(s), Q_h^{\lambda,\sigma}(s, \bullet, \pi, \mu)\right\rangle\right]$$
$$= J^{\lambda,\sigma}(\mu, \pi) - J^{\lambda,\sigma}(\mu, \widetilde{\pi}) - \lambda D_{m[\widetilde{\pi}]}(\widetilde{\pi}, \sigma) + \lambda D_{m[\widetilde{\pi}]}(\pi, \sigma),$$

*where we set $\mu = m[\pi]$.*

**Proof.** From the definition of $V^{\lambda,\sigma}$ and $Q^{\lambda,\sigma}$ in (4.2) and (4.3), we have

$$\sum_{h=1}^{H} \mathbb{E}_{s\sim m[\widetilde{\pi}]_h} \left[ \left\langle \pi_h(s), Q_h^{\lambda,\sigma}(s,\bullet,\pi,\mu) \right\rangle \right]$$

$$= \sum_{h=1}^{H} \mathbb{E}_{s\sim m[\widetilde{\pi}]_h} \left[ \left\langle \pi_h(s), r_h(s,\bullet,\mu_h) + \mathbb{E}\left[ V_{h+1}^{\lambda,\sigma}(s_{h+1},\mu,\pi) \mid s_{h+1}\sim P(s,\bullet,\mu_h) \right] \right\rangle \right]$$

$$= \sum_{h=1}^{H} \mathbb{E}_{s_h\sim m[\widetilde{\pi}]_h} \left[ \mathbb{E}_{a_h\sim\pi_h(s)} \left[ r_h(s_h,a_h,\mu_h) - \lambda D_{\mathrm{KL}}(\pi(s_h),\sigma(s_h)) \right] \right] + \lambda D_{m[\widetilde{\pi}]}(\pi,\sigma)$$

$$+ \sum_{h=1}^{H} \mathbb{E}_{s\sim m[\widetilde{\pi}]_h} \left[ \mathbb{E}\left[ V_{h+1}^{\lambda,\sigma}(s_{h+1},\mu,\pi) \mid s_{h+1}\sim P(s,a_h,\mu_h), a_h\sim\pi_h(s) \right] \right] \tag{E.2}$$

$$= \sum_{h=1}^{H} \mathbb{E}_{s_h\sim m[\widetilde{\pi}]_h} \left[ V_h^{\lambda,\sigma}(s_h,\mu,\pi) - \mathbb{E}\left[ V_{h+1}^{\lambda,\sigma}(s_{h+1},\mu,\pi) \; \middle| \; \begin{array}{c} s_{h+1}\sim P(s,a_h,\mu_h), \\ a_h\sim\pi_h(s) \end{array} \right] \right]$$

$$+ \lambda D_{m[\widetilde{\pi}]}(\pi,\sigma) + \sum_{h=1}^{H} \mathbb{E}_{s\sim m[\widetilde{\pi}]_h} \left[ \mathbb{E}\left[ V_{h+1}^{\lambda,\sigma}(s_{h+1},\mu,\pi) \; \middle| \; \begin{array}{c} s_{h+1}\sim P(s,a_h,\mu_h), \\ a_h\sim\pi_h(s) \end{array} \right] \right]$$

$$= \sum_{h=1}^{H} \mathbb{E}_{s\sim m[\widetilde{\pi}]_h} \left[ V_h^{\lambda,\sigma}(s,\mu,\pi) \right] + \lambda D_{m[\widetilde{\pi}]}(\pi,\sigma).$$

Similarly, (4.1) and (2.1) gives us

$$\sum_{h=1}^{H} \mathbb{E}_{s\sim m[\widetilde{\pi}]_h} \left[ \left\langle \widetilde{\pi}_h(s), Q_h^{\lambda,\sigma}(s,\bullet,\pi,\mu) \right\rangle \right]$$

$$= \sum_{h=1}^{H} \mathbb{E}_{s_h\sim m[\widetilde{\pi}]_h} \left[ \mathbb{E}_{a_h\sim\widetilde{\pi}_h(s)} \left[ r_h(s_h,a_h,\mu_h) - \lambda D_{\mathrm{KL}}(\widetilde{\pi}(s_h),\sigma(s_h)) \right] \right] + \lambda D_{m[\widetilde{\pi}]}(\widetilde{\pi},\sigma)$$

$$+ \sum_{h=1}^{H} \mathbb{E}_{s\sim m[\widetilde{\pi}]_h} \left[ \mathbb{E}\left[ V_{h+1}^{\lambda,\sigma}(s_{h+1},\mu,\pi) \mid s_{h+1}\sim P(s,a_h,\mu_h), a_h\sim\widetilde{\pi}_h(s) \right] \right] \tag{E.3}$$

$$= J^{\lambda,\sigma}(\mu,\widetilde{\pi}) + \lambda D_{m[\widetilde{\pi}]}(\widetilde{\pi},\sigma) + \sum_{h=1}^{H} \mathbb{E}_{s\sim m[\widetilde{\pi}]_{h+1}} \left[ V_{h+1}^{\lambda,\sigma}(s,\mu,\pi) \right].$$

Combining (E.2) and (E.3) yields

$$\sum_{h=1}^{H} \mathbb{E}_{s\sim m[\widetilde{\mu}]_h} \left[ \left\langle (\pi_h - \widetilde{\pi}_h)(s), Q_h^{\lambda,\sigma}(s,\bullet,\pi,\mu) \right\rangle \right]$$

$$= \left( \sum_{h=1}^{H} \mathbb{E}_{s\sim m[\widetilde{\pi}]_h} \left[ V_h^{\lambda,\sigma}(s,\mu,\pi) \right] + \lambda D_{m[\widetilde{\pi}]}(\pi,\sigma) \right)$$

$$- \left( J^{\lambda,\sigma}(\mu,\widetilde{\pi}) + \lambda D_{m[\widetilde{\pi}]}(\widetilde{\pi},\sigma) + \sum_{h=1}^{H} \mathbb{E}_{s\sim m[\widetilde{\pi}]_{h+1}} \left[ V_{h+1}^{\lambda,\sigma}(s,\mu,\pi) \right] \right)$$

$$= \left( \mathbb{E}_{s\sim m[\widetilde{\pi}]_1} \left[ V_1^{\lambda,\sigma}(s,\mu,\pi) \right] + \lambda D_{m[\widetilde{\pi}]}(\pi,\sigma) \right) - \left( J^{\lambda,\sigma}(\mu,\widetilde{\pi}) + \lambda D_{m[\widetilde{\pi}]}(\widetilde{\pi},\sigma) \right)$$

$$= \mathbb{E}_{s\sim\mu_1} \left[ V_1^{\lambda,\sigma}(s,\mu,\pi) \right] - J^{\lambda,\sigma}(\mu,\widetilde{\pi}) + \lambda D_{m[\widetilde{\pi}]}(\pi,\sigma) - \lambda D_{m[\widetilde{\pi}]}(\widetilde{\pi},\sigma),$$

which concludes the proof. ∎

**Lemma E.5.** *For all $\pi, \widetilde{\pi} \in (\Delta(\mathcal{A})^{\mathcal{S}})^H$, it holds that*

$$\sum_{h=1}^{H} \mathbb{E}_{s\sim m[\widetilde{\pi}]_h} \left[ \left\langle (\pi_h - \widetilde{\pi}_h)(s), \log\frac{\pi_h(s)}{\sigma_h(s)} \right\rangle \right] = D_{m[\widetilde{\pi}]}(\pi,\sigma) - D_{m[\widetilde{\pi}]}(\widetilde{\pi},\sigma) + D_{\widetilde{\pi}}(\widetilde{\pi},\pi).$$

*Proof.* A direct computation yields

$$\sum_{h=1}^{H} \mathbb{E}_{s \sim m[\widetilde{\pi}]_h} \left[ \left\langle (\pi_h - \widetilde{\pi}_h)(s), \log \frac{\pi_h(s)}{\sigma_h(s)} \right\rangle \right]$$

$$= D_{m[\widetilde{\pi}]}(\pi, \sigma) - \sum_{h=1}^{H} \mathbb{E}_{s \sim m[\widetilde{\pi}]_h} \left[ \left\langle \widetilde{\pi}_h(s), \log \frac{\widetilde{\pi}_h(s)}{\sigma_h(s)} - \log \frac{\widetilde{\pi}(s)}{\pi(s)} \right\rangle \right]$$

$$= D_{m[\widetilde{\pi}]}(\pi, \sigma) - D_{m[\widetilde{\pi}]}(\widetilde{\pi}, \sigma) + D_{m[\widetilde{\pi}]}(\widetilde{\pi}, \pi).$$

∎

**Lemma E.6.** *The operator $m$ defined in* (2.1) *is 1-Lipschitz, namely, it holds that*

$$\|m[\pi]_{h+1} - m[\pi']_{h+1}\| \leq \sum_{l=0}^{h} \mathbb{E}_{s_l \sim m[\pi]_l} [\|\pi_l(s_l) - \pi'_l(s_l)\|], \tag{E.4}$$

*for $\pi, \pi' \in (\Delta(\mathcal{A})^{\mathcal{S}})^H$ and all $h \in \{0, \dots, H\}$. Here, we set $\pi_0(s) = \pi'_0(s) = U_{\mathcal{A}}$ for all $s \in \mathcal{S}$.*

*Proof.* Fix $\pi, \pi' \in (\Delta(\mathcal{A})^{\mathcal{S}})^H$. We prove the inequality by induction on $h$.

**(I) Base step $h = 0$:** It is obvious because $\|m[\pi]_1 - m[\pi']_1\| = \|\mu_1 - \mu_1\| = 0$.

**(II) Inductive step:** Suppose that there exists $h \in [H]$ satisfying the inequality (E.4). By (2.1), we obtain

$$\|m[\pi]_{h+2} - m[\pi']_{h+2}\|$$

$$\leq \sum_{\substack{s_{h+2} \in \mathcal{S}, \\ (s_{h+1}, a_{h+1}) \in \mathcal{S} \times \mathcal{A}}} P_{h+1}(s_{h+2} \mid s_{h+1}, a_{h+1}) m[\pi]_{h+1}(s_{h+1}) \big| \pi_{h+1}(a_{h+1} \mid s_{h+1}) - \pi'_{h+1}(a_{h+1} \mid s_{h+1}) \big|$$

$$+ \sum_{\substack{s_{h+2} \in \mathcal{S}, \\ (s_{h+1}, a_{h+1}) \in \mathcal{S} \times \mathcal{A}}} P_{h+1}(s_{h+2} \mid s_{h+1}, a_{h+1}) \pi'_{h+1}(a_{h+1} \mid s_{h+1}) |m[\pi]_{h+1}(s_{h+1}) - m[\pi']_{h+1}(s_{h+1})|$$

$$\leq \sum_{(s_{h+1}, a_{h+1}) \in \mathcal{S} \times \mathcal{A}} m[\pi]_{h+1}(s_{h+1}) \big| \pi_{h+1}(a_{h+1} \mid s_{h+1}) - \pi'_{h+1}(a_{h+1} \mid s_{h+1}) \big|$$

$$+ \sum_{s_{h+1} \in \mathcal{S}} |m[\pi]_{h+1}(s_{h+1}) - m[\pi']_{h+1}(s_{h+1})|$$

$$= \mathbb{E}_{s_{h+1} \sim m[\pi]_{h+1}} \left[ \big\| \pi_{h+1}(s_{h+1}) - \pi'_{h+1}(s_{h+1}) \big\| \right] + \|m[\pi]_{h+1} - m[\pi']_{h+1}\|.$$

By the hypothesis of the induction, we finally obtain

$$\|m[\pi]_{h+2} - m[\pi']_{h+2}\|$$

$$\leq \mathbb{E}_{s \sim m[\pi]_{h+1}} \left[ \big\| \pi_{h+1}(s) - \pi'_{h+1}(s) \big\| \right] + \sum_{l=1}^{h} \mathbb{E}_{s \sim m[\pi]_l} \|\pi_l(s) - \pi'_l(s)\|$$

$$\leq \sum_{l=1}^{h+1} \mathbb{E}_{s \sim m[\pi]_l} \|\pi_l(s) - \pi'_l(s)\|.$$

∎

**Lemma E.7.** *Let $\pi, \pi' \in (\Delta(\mathcal{A})^{\mathcal{S}})^H$, $\mu, \mu' \in \Delta(\mathcal{S})^H$, $s \in \mathcal{S}$, and $h \in \{1, \dots, H+1\}$. Assume*

$$\min_{(h,a,s) \in [H] \times \mathcal{A} \times \mathcal{S}} \min\{\pi_h(a \mid s), \pi'_h(a \mid s)\} > 0,$$

*and set $\mu_{H+1} = \mu'_{H+1} = U_{\mathcal{S}}$, $\pi_{H+1}(s) = \pi'_{H+1}(s) = U_{\mathcal{A}}$ for all $s \in \mathcal{S}$.*

$$\left| V_h^{\lambda,\sigma}(s, \pi, \mu) - V_h^{\lambda,\sigma}(s, \pi', \mu') \right|$$

$$\leq \mathbb{E} \left[ \sum_{l=h}^{H+1} \left( C^{\lambda,\sigma}(\pi, \pi') \|\pi_l(s_l) - \pi'_l(s_l)\|_1 + L\|\mu_l - \mu'_l\|_1 \right) \middle| \begin{array}{c} s_h = s, \\ s_{l+1} \sim P_l(s_l, a_l), \\ a_l \sim \pi_l(s_l) \\ \text{for each } l \in \{h, \dots, H+1\} \end{array} \right]$$

*for Here, $C^{\lambda,\sigma}(\pi,\pi') > 0$ is defined in [Proposition E.8], and the discrete time stochastic process $(s_l)_{l=h}^{H}$ is induced recursively as $s_{l+1} \sim P_l(s_l, a_l), a_l \sim \pi_l(s_l)$ for each $l \in \{h, \ldots, H-1\}$.*

**Proof.** Fix $\pi$, $\pi'$, $\mu$ and $\mu'$. We prove the inequality by backward induction on $h$.

**(I) Base step $h = H + 1$:** It is obvious because $\left| V_{H+1}^{\lambda,\sigma}(s, \pi, \mu) - V_{H+1}^{\lambda,\sigma}(s, \pi', \mu') \right| = |0 - 0| = 0$.

**(II) Inductive step:** Suppose that there exists $h \in [H]$ satisfying

$$\left| V_{h+1}^{\lambda,\sigma}(s, \pi, \mu) - V_{h+1}^{\lambda,\sigma}(s, \pi', \mu') \right|$$
$$\leq \mathbb{E}\left[ \sum_{l=h+1}^{H+1} \left( C^{\lambda,\sigma}(\pi,\pi') \|\pi_l(s_l) - \pi_l'(s_l)\|_1 + L\|\mu_h - \mu_h'\|_1 \right) \, \middle| \, \begin{array}{l} s_{h+1} = s, \\ s_{l+1} \sim P_l(s_l, a_l), \\ a_l \sim \pi_l(s_l) \\ \text{for each } l \in \{h+1, \ldots, H+1\} \end{array} \right],$$
(E.5)

for all $s \in \mathcal{S}$. By the definition of the value function in (4.2) and [Assumption 2.5], we have

$$\left| V_h^{\lambda,\sigma}(s, \pi, \mu) - V_h^{\lambda,\sigma}(s, \pi', \mu') \right|$$

$$\leq \left| \sum_{a_h \in \mathcal{A}} \left( \pi_h(a_h \mid s) r_h(s, a_h, \mu_h) - \pi_h'(a_h \mid s) r_h(s, a_h, \mu_h') \right) \right|$$

$$+ \lambda |D_{\mathrm{KL}}(\pi_h(s), \sigma_h(s)) - D_{\mathrm{KL}}(\pi_h'(s), \sigma_h(s))|$$

$$+ \left| \sum_{\substack{a_h \in \mathcal{A}, \\ s_{h+1} \in \mathcal{S}}} P_h(s_{h+1} \mid s, a_h) \left( \pi_h(a_h \mid s) V_{h+1}^{\lambda,\sigma}(s_{h+1}, \pi, \mu) - \pi'(a_h \mid s) V_{h+1}^{\lambda,\sigma}(s_{h+1}, \pi', \mu') \right) \right|$$

$$\leq \|\pi_h(s) - \pi_h'(s)\|_1 + \sum_{a_h \in \mathcal{A}} \pi_h(a_h \mid s) |r_h(s, a_h, \mu_h) - r_h(s, a_h, \mu_h')|$$

$$+ \lambda \left| \sum_{a_h \in \mathcal{A}} \left( \pi_h(a_h \mid s) \left( \log \frac{\pi_h(a_h \mid s)}{\sigma_h(a_h \mid s)} - 1 \right) - \pi_h'(a_h \mid s) \left( \log \frac{\pi_h'(a_h \mid s)}{\sigma_h(a_h \mid s)} - 1 \right) \right) \right|$$

$$+ \|\pi_h(s) - \pi_h'(s)\|_1$$

$$+ \sum_{\substack{a_h \in \mathcal{A}, \\ s_{h+1} \in \mathcal{S}}} P_h(s_{h+1} \mid s, a_h) \pi_h(a_h \mid s) \left| V_{h+1}^{\lambda,\sigma}(s_{h+1}, \pi, \mu) - V_{h+1}^{\lambda,\sigma}(s_{h+1}, \pi', \mu') \right|$$

$$\leq 2\|\pi_h(s) - \pi_h'(s)\|_1 + L\|\mu_h - \mu_h'\|_1$$

$$+ \lambda \max_{(h,a,s)} \log \frac{1}{(\sigma\pi\pi')_h(a \mid s)} \|\pi_h(s) - \pi_h'(s)\|_1$$

$$+ \sum_{\substack{a_h \in \mathcal{A}, \\ s_{h+1} \in \mathcal{S}}} P_h(s_{h+1} \mid s, a_h) \pi_h(a_h \mid s) \left| V_{h+1}^{\lambda,\sigma}(s_{h+1}, \pi, \mu) - V_{h+1}^{\lambda,\sigma}(s_{h+1}, \pi', \mu') \right|$$

$$\leq C^{\lambda,\sigma}(\pi,\pi') \|\pi_h(s) - \pi_h'(s)\|_1 + L\|\mu_h - \mu_h'\|_1$$

$$+ \mathbb{E}\left[ \left| V_{h+1}^{\lambda,\sigma}(s_{h+1}, \pi, \mu) - V_{h+1}^{\lambda,\sigma}(s_{h+1}, \pi', \mu') \right| \, \middle| \, \begin{array}{l} s_h = s, \\ s_{h+1} \sim P_h(s_h, a_h), \\ a_h \sim \pi_h(s_h) \end{array} \right].$$

Combining the above inequality and the hypothesis of the induction completes the proof. ∎

**Proposition E.8.** *Let $Q^{\lambda,\sigma}$ be the function defined by (4.3), and $(\pi, \pi') \in \left( (\Delta(\mathcal{A})^{\mathcal{S}})^H \right)^2$ be policies with full supports. Under Assumptions 2.5 and 4.1, it holds that*

$$\left| Q_h^{\lambda,\sigma}(s, a, \pi, \mu) - Q_h^{\lambda,\sigma}(s, a, \pi', \mu') \right|$$

$$\leq L \sum_{l=h}^{H} \| \mu_l - \mu'_l \| + C^{\lambda,\sigma}(\pi, \pi') \, \mathbb{E}_{(s_l)_{l=h+1}^H} \left[ \sum_{l=h+1}^{H} \| \pi_l(s_l) - \pi'_l(s_l) \| \; \middle| \; s_h = s \right],$$

*for $(h, s, a) \in [H] \times \mathcal{S} \times \mathcal{A}$ and $\mu, \mu' \in \Delta(\mathcal{S})^H$. Here, the random variables $(s_l)_{l=h+1}^H$ follows the stochastic process starting from state $s$ at time $h$, induced from $P$ and $\pi$, and the function $C^{\lambda,\sigma}\colon \left( (\Delta(\mathcal{A})^{\mathcal{S}})^H \right)^2 \to \mathbb{R}$ is given by $C^{\lambda,\sigma}(\pi, \pi') = 2 - \lambda \inf_{(h,s,a) \in [H] \times \mathcal{S} \times \mathcal{A}} \log (\sigma \pi \pi')_h (a \mid s)$.*

***Proof of Proposition E.8.*** Let $h$ be larger than 2. By the definition of $Q_h^{\lambda,\sigma}$ given in (4.3) and Lemma E.7, we have

$$\left| Q_{h-1}^{\lambda,\sigma}(s, a, \pi, \mu) - Q_{h-1}^{\lambda,\sigma}(s, a, \pi', \mu') \right|$$

$$\leq \left| r_{h-1}(s, a, \mu_{h-1}) - r_{h-1}(s, a, \mu'_{h-1}) \right| + \mathbb{E}_{s_h \sim P_{h-1}(s,a)} \left[ \left| V_h^{\lambda,\sigma}(s_h, \pi, \mu) - V_h^{\lambda,\sigma}(s_h, \pi', \mu') \right| \right]$$

$$\leq L \| \mu_{h-1} - \mu'_{h-1} \| + \mathbb{E}_{s_h \sim P_{h-1}(s,a)} \left[ \left| V_h^{\lambda,\sigma}(s_h, \pi, \mu) - V_h^{\lambda,\sigma}(s_h, \pi', \mu') \right| \right].$$

Combining the above inequality and Lemma E.7 completes the proof. ∎

# F   Experiment Details

We ran experiments on a laptop with an 11th Gen Intel Core i7-1165G7 8-core CPU, 16GB RAM, running Windows 11 Pro with WSL. As is clear from Algorithm 1, APP is deterministic. Thus, we ran the algorithm only once for each experimental setting. We implemented APP using Python. The computation of $Q^{\lambda,\sigma}$ and $\mu$ in Algorithm 1 was based on the implementation provided by Fabian et al. (2023).

**Algorithms.**   In this experiment, we implement APP in Algorithm 1. For comparison, we also implement RMD (i.e., Algorithm 1 without the update of $\sigma_k$) in (4.4). For both algorithms, the learning rate is fixed at $\eta = 0.1$, and we vary the regularization parameter $\lambda$ and update time $T$ to run the experiments.

We show further details for the Beach Bar Process. We set $H = 10, |\mathcal{S}| = 10, \mathcal{A} = \{-1, \pm 0, +1\}, \lambda = 0.1, \eta = 0.1$, and

$$P_h(s' \mid s, a) = \begin{cases} 1 - \varepsilon & \text{if } a = \pm 0 \text{ \& } s' = s, \\ \dfrac{\varepsilon}{2} & \text{if } a = \pm 1 \text{ \& } s' = s \pm 1, \\ 0 & \text{otherwise,} \end{cases}$$

where we choose $\varepsilon = 0.1$. In addition, we initialize $\sigma^0$ and $\pi^0$ in Algorithm 1 as the uniform distributions on $\mathcal{A}$.

*Remark F.1.* When the contraction factor $1 - \lambda\eta$ is close to 1, we can observe a small $\tau$ can lead to instability in the outer PP loop.

