# OpenReview forum: "Last Iterate Convergence in Monotone Mean Field Games"
_NeurIPS.cc/2025/Conference — NeurIPS 2025 poster_

### Official Review · Reviewer_z4KT · 2025-06-23

**Clarity:** 2
**Significance:** 2
**Originality:** 3
**Rating:** 4
**Confidence:** 3

**Summary:**

The authors consider an algorithm for learning equilibria in mean field games with last iterate convergence, to obtain low exploitability during final stages of learning, as opposed to playing suboptimal during learning in methods such as fictitious play. The algorithm is supported by rigorous theoretical analysis of convergence, as well as empirically by performing a successful numerical experiment.

**Questions:**

- Where does Figure 1 come from? Is it from the experiment further below?
- I think the explanation on line 203 is unclear, how does Lemma E.7 provide the upper bound of exploitability? Can this be made more formal?
- The convergence result in Thm. 4.4 given in $D_{\mu^*}$, how does this imply exponential convergence in a standard metric for the policy, or in exploitability?
- Why is (3.1) called intractable? If it is just a regularized MFG, can you not solve it using any regularized MFG algorithm?

**Ethical Concerns:**

["NO or VERY MINOR ethics concerns only"]

**Final Justification:**

I appreciate the additional explanations and would like to keep the score, as all the questions have been answered.

**Limitations:**

yes

**Paper Formatting Concerns:**

The references seem to not follow the right format.

**Quality:**

3

**Strengths And Weaknesses:**

The claims of last-iterate convergence are supported by theoretical results, as well as experimentally in one example problem. The benchmark Beach Bar problem makes sense for the field of learning MF equilibria. The experiments are limited and could be extended to show the limitations of the proposed methodology beyond the Beach Bar example, beyond regularized MFGs or even beyond the theoretical assumptions made in the work. In particular, there is no MF interaction in the dynamics, which is a common theoretical assumption but might be circumvented in practice / experiments.

The contributions would seem relevant, to consider last iterate convergence in more general monotone settings, as opposed to existing works that require averaging after playing as in fictitious play or considering special cases such as graphon MFGs. However,  without application to a data-driven setting with N-agent systems and without access to the model, it seems although LIC is achieved, model knowledge and exact calculation of Q-functions is necessary. In that case, I am not sure of the advantage of LIC over any other convergence in existing literature, which could be compared more clearly.

The clarity could be improved by stating concisely the differences to existing theoretical results and algorithms, whenever a new result is introduced, e.g., Thm. 4.3 and Thm 4.4. Some things remained unclear for me, see the questions section below.

Minor things:
- I think [1] is also relevant to add to Table 1, as it has discrete-time convergence results for MFGs (under concavity assumptions)
- Typo line 671: "for Here"

[1] Geist, Matthieu, et al. "Concave Utility Reinforcement Learning: The Mean-field Game Viewpoint." Proceedings of the 21st International Conference on Autonomous Agents and Multiagent Systems. 2022.

---

> ### Author Rebuttal · Authors · 2025-07-30
>
> Thank you for your careful review and for your interest in our convergence guarantees for MFGs. We address your comments below.
>
> >Minor things:
> > I think [1] is also relevant to add to Table 1, as it has discrete-time convergence results for MFGs (under concavity assumptions)
> > Typo line 671: “for Here”
>
> Thank you for pointing these out. We will add [1] to Table 1 and correct the typo in the revision.
>
> > Where does Figure 1 come from? Is it from the experiment further below?
>
> It is mentioned briefly in line 169. Figure 1 illustrates our experiments showing that Regularized Mirror Descent converges exponentially in practice.
> > how does Lemma E.7 provide the upper bound of exploitability? Can this be made more formal?
> > The convergence result in Thm. 4.4 given in $D_{\mu^\ast}$, how does this imply exponential convergence in a standard metric for the policy, or in exploitability?
>
> Thank you for this request. We shall explain how to get the bound in detail.
> From Lemma E.7 and the identity $J(\mu,\pi)=\mathbb{E}\_{s\sim\mu\_1}[V_1(s,\mu,\pi)]$, we can bound $$\operatorname{Exploit}(\pi^t)\le J(m[\varpi^\ast],\pi_{\textup{BR}}^t)-J(m[\varpi^\ast],\pi^t)+\mathcal{O}(\\|m[\varpi^\ast]-m[\pi^t]\\|),$$ where $\pi^t_{\textup{BR}}$ is a maximizer of $J(m[\pi^t],\cdot)$.
>
> Furthermore, by the definition of equilibrium, $J(m[\varpi^\ast],\pi_{\textup{BR}}^t)\le J(m[\varpi^\ast],\varpi^\ast)$, so applying Lemma E.7 again yields $$\operatorname{Exploit}(\pi^t)=\mathcal{O}(\\|m[\varpi^\ast]-m[\pi^t]\\|+\sum_h\mathbb{E}\_{s\sim\mu_1}[\\|\varpi^\ast_h(s)-\pi^t_h(s)\\|]).$$
>
> Finally, using Pinsker’s inequality and Lemma E.6 (l.631), we obtain $$\operatorname{Exploit}(\pi^t)=\mathcal{O}(\sqrt{D_{\mu^\ast}(\varpi^\ast,\pi^t)}).$$
>
> > Why is (3.1) called intractable? If it is just a regularized MFG, can you not solve it using any regularized MFG algorithm?
>
> “Intractable” here means that finding the pair $(\sigma^{k+1},\pi^{k+1})$ in equation (3.1) exactly is difficult. Most regularized MFG algorithms—including RMD—can only compute this pair approximately. While one could solve a linear program to obtain an exact solution, doing so is not generally scalable. Note that RMD and APP are scalable methods when combined with function approximation.

---

> > ### Comment · Reviewer_z4KT · 2025-08-04
> >
> > Thank you for the clarifications! I appreciate the additional explanations and would like to keep the score.

---

> > > ### Author Response · Authors · 2025-08-05
> > >
> > > We appreciate your understanding. We would like to reflect the added explanation in the revised manuscript.

---

### Official Review · Reviewer_GbwY · 2025-06-29

**Clarity:** 3
**Significance:** 2
**Originality:** 2
**Rating:** 4
**Confidence:** 4

**Summary:**

The paper addresses the challenge of achieving last-iterate convergence (LIC) in Mean-Field Games (MFGs), especially under the more realistic assumption of non-strict monotonicity. The last-iterated policy of a proximal-point (PP) update, which incorporates Kullback–Leibler (KL) regularization, is shown to converge to an MFG equilibrium even under non-strict monotonicity. Each PP update step is shown to be equivalent to finding the equilibria of a KL-regularized MFG, and this regularized equilibrium can be found efficiently using Regularized Mirror Descent (RMD) with an exponential last-iterate convergence rate. Building on these theoretical insights, the paper proposes the Approximate Proximal-Point (APP) algorithm. APP approximates the PP update by executing a small number of RMD steps. Numerical experiments on standard benchmarks, such as the Beach Bar Process, show that APP reliably converges to the un-regularized mean-field equilibrium without time-averaging.

**Questions:**

1. How do the "optimization-based methods for MFGs" (e.g., Guo et al. 2024) that achieve local convergence without monotonicity compare in practical utility to the "global" convergence results achieved here with monotonicity?
2. The assumption of ergodicity is too strong. Zaman et al. 2023, for example, establish results under weaker class of communicating MDPs. How would the results presented in the paper generalize to such MDPs?
3. The assumption of mean-field independent MDP is also strong for MFGs (Zaman et.al 2023, Zeng et.al 2024, etc.) consider the case where $P$ depends on $\mu$. How easy is it to relax this assumption in the paper?
4. Can the authors elaborate why the initial policy~$\pi_0$ should have full support in Prop. C.1?

**Ethical Concerns:**

["NO or VERY MINOR ethics concerns only"]

**Final Justification:**

The rebuttal helped clarify the key concerns.

**Limitations:**

Yes

**Quality:**

2

**Strengths And Weaknesses:**

Strengths:
1. The paper directly tackles the crucial problem of LIC in MFGs, which is relevant for deep learning applications where time-averaged policies can be less meaningful due to nonlinearity in parameter space.
2. Establishing convergence under non-strict monotonicity enhances the applicability in practice.
3. Establishing exponential convergence rate for RMD is a useful result, improving previous results that achieved only polynomial or time-averaged convergence.
4. The Proximal-Point (PP) method, with un-linearized objective, is expected to be less sensitive to approximation error and more robust in non-strictly monotone settings compared to standard Mirror Descent (MD).
5. Proposition C.1 is interesting.

Weaknesses:
1. The model is assumed to be known, and naturally no sample complexity analyses like the most recent references. A more realistic setting would have been the case where the parameters are learned from the data.
2. A couple of very strong assumptions in relation to the latest literature on MFGs (see Questions)
3. It is not clear how the results are placed in the literature when considered in conjunction with the rather `outdated' assumptions used.

---

> ### Author Rebuttal · Authors · 2025-07-30
>
> Thank you for reviewing our paper. We appreciate the opportunity to clarify several points where our exposition may have fallen short regarding our setting and its differences from other works. Below, we explain the position of our paper in this context.
> > How do the "optimization-based methods for MFGs" (e.g., Guo et al. 2024) that achieve local convergence without monotonicity compare in practical utility to the "global" convergence results achieved here with monotonicity?
>
> Local‑convergence guaranteed methods require the initial policy $\pi^0$ to be sufficiently close to an equilibrium—a condition that cannot be verified a priori—and impose no structural assumptions on the MFG. By contrast, our “global” convergence guarantees require only monotonicity and full support of $\pi^0$, which can be checked before running the algorithm, to ensure convergence of PP and RMD.
>
> In addition, Online Mirror descent-based methods, including RMD, can be scaled to large MFGs using function approximation with deep learning, but "optimization-based methods for MFGs “ would be difficult to scale in a similar way.
>
> > The assumption of ergodicity is too strong. Zaman et al. 2023, for example, establish results under weaker class of communicating MDPs. How would the results presented in the paper generalize to such MDPs?
>
> We do not assume the ergodicity (i.e., that any state is reachable from **every** other state under **some** policy). Instead, Assumption 2.1 requires only that any state be reachable from **some** states under **any** policy. This condition also subsumes the communicating‑MDP setting of Zaman et al. (2023).
>
> > The assumption of mean-field independent MDP is also strong for MFGs (Zaman et al. 2023, Zeng et al. 2024, etc.) consider the case where $P$ depends on $\mu$. How easy is it to relax this assumption in the paper?
>
> First, Zaman et al. and Zeng et al. impose different conditions on the transition dynamics, such as the contraction assumption (Assumption 1 in Zaman et al.) and the herding condition (Assumption 4 in Zeng et al.), which differ from our monotonicity assumption. Still, neither is strictly stronger than the other. Monotonicity corresponds to convexity in optimization and arises naturally in applications like congestion modeling, whereas a contraction assumption is comparatively strong and imposed purely for theoretical convergence. The herding condition is used where the uniqueness of the equilibrium need not always hold; under our non‑strict monotonicity assumption, the uniqueness of the (unregularized) equilibrium also does not generally follow.
>
> Extending convergence of MD‑based algorithms to mean‑field dependent transitions would require fundamentally different proof techniques—for example, one would need an alternative to Lemma E.4 implicitly used in Perolat et al. (2022) and F. Zhang et al. (2023).
>
> > Can the authors elaborate why the initial policy $\pi^0$ should have full support in Prop. C.1?
>
> Full support was not required in Proposition C.1—this result does not depend on $\pi^0.$ We will write this point more clearly in the revision.

---

### Official Review · Reviewer_YWXi · 2025-07-01

**Clarity:** 3
**Significance:** 2
**Originality:** 2
**Rating:** 3
**Confidence:** 3

**Summary:**

This paper studies a monotone mean field game without strict monotonicity. It proposes a proximal point-like algorithm that can solve the regularized version of the MFG exponentially fast. Numerical experiments also validate the proposed algorithm empirically.

**Questions:**

Could you explain the technical novelty of the analysis? It seems like the improvement over Zhang et al is on the non-strict monotonicity assumption. But Dong et al's results are also under non-strict monotonicity.

Could the poly-> exp improvement on the convergence rate be attributed to different feedback types as Dong et al focused on bandit feedback?

**Ethical Concerns:**

["NO or VERY MINOR ethics concerns only"]

**Final Justification:**

After the reviewer-author discussion, I do agree that the analysis and result, although expected, is new. I don't oppose to the acceptance of this paper.

**Limitations:**

Yes

**Quality:**

3

**Strengths And Weaknesses:**

The paper is nicely presented with sufficient explanation and insights into how the algorithm is developed.

Strength: the proximal point like algorithm is very well explained with figures to aid the understanding of it. The intuition from the continuous time version of the algorithm also help motivates and bridge the connection to previous work on continuous time mean field algorithms.

Weakness: My main concern is that the exponential convergence rate is not new to regularized games (see Cen et al 2023, Zhang et al 2023, Dong et al 2025). To my knowledge, these papers also use a proximal point-like algorithm. From my brief skim of the proof, I cannot tell the technical novelty of the analysis and I am not sure if it could be straightforwardly extended from the existing analysis.

I am not fully convinced by the contribution of this work because the proposed proximal-point–like algorithm does not appear significantly different from the extragradient-type algorithm used in Cen et al. (2023). Although the authors argue that the main challenge in extending the analysis of Cen et al. (2023) lies in the backward induction of the Q-function (Remark 4.6), this issue is already well-known in monotone mean-field games and has been addressed in Zhang et al. 2023 (Section F) and Dong et al. 2025 (Lemma B.1).

One key difference between the two work lies on that Dong et al uses the naive mirror descent algorithm, while this one uses the proximal point like algorithm. Combining bandit feedback is difficult with the proximal point algorithm, so I would have expected that the improvement on the convergence speed stem from here. Similar reasoning can be applied to Zhang et al. One improvement of the results in Theorem 4.3 is that it improves the terms in Equation E.1 that are due to the exploration with bandit feedback (Lemma I.3).
But we only care about the full information feedback, the exponential speedup brought by the proximal point like algorithm, combined with the speed up from not having to explore,  is not too surprising an this combination is known to be able to do that in regular non-mean-field monotone game (Cen et al).

---

> ### Author Rebuttal · Authors · 2025-07-30
>
> Thank you for your feedback.
> First, we apologize for a typo in Line 71 of the Contributions box. It should read *Theorem 3.1* and *Theorem 4.3* (not 4.3 and 4.4).
> We answer the following comment and question:
>
> > My main concern is that the exponential convergence rate is not new to regularized games (see Cen et al 2023, Zhang et al 2023, Dong et al 2025). To my knowledge, these papers also use a proximal point-like algorithm.  To my knowledge, these papers also use a proximal point-like algorithm.
>
>
> The exponential convergence is a novel result for the regularized Mean-Field Game. Cen et al.'s result is not for MFG, and Zhang et al. Dong et al. only achieve polynomial rates. The methods in these previous studies are results for policy gradient and mirror descent, which are different from our proximal point-type algorithm (3.1).
>
>
>
>
> > Could you explain the technical novelty of the analysis? It seems like the improvement over Zhang et al is on the non-strict monotonicity assumption. But Dong et al’s results are also under non-strict monotonicity.
>
> First, we would like to emphasize that our key technical novelty is not only the last-iterate convergence (LIC) for *regularized* MFGs, but also the LIC result for **unregularized** MFGs (Theorem 3.1). To our knowledge, this is the first guarantee of LIC in the unregularized setting. Under unregularized dynamics, no prior work has established LIC.
> Furthermore, our LIC result for the regularized problem (Theorem 4.3) strictly improves upon Zhang et al.'s and Dong's full-feedback analyses by establishing an asymptotically faster convergence rate under non-strict monotonicity.
> The key to establishing these two theoretical contributions is correctly evaluating the change in $\mu$. In Theorem 3.1, we were able to neglect $\mu^{k+1}$ of $D_{\mu^{k+1}}$ using Łojasiewicz's inequality; in the proof of Theorem 4.6, we evaluate the dependence of Q-function on $\mu^{t}$ by $D_{\mu^\ast}(\varpi^\ast,\pi^t)$ using the KL divergence property. See Remark 3.3 and 4.6 for details.
>
>
> > Could the poly→exp improvement on the convergence rate be attributed to different feedback types as Dong et al focused on bandit feedback?
>
>
>
> No—Zhang et al.’s convergence rate remains polynomial even under full feedback, whereas our analysis yields a strictly faster (exponential-phase) rate under full feedback. Moreover, their bandit-feedback results build directly on their full-feedback polynomial convergence (Theorem 5.1), so they only achieve a polynomial rate in the bandit setting. Therefore, using our exponential-rate guarantee as a foundation, one can expect improved convergence rates under bandit feedback.
> Although extending our contraction-based analysis to the bandit setting is non-trivial, we believe our techniques could be used to derive stronger polynomial (or even exponential-phase) rates for bandit-feedback algorithms in MFGs, and we plan to explore this in future work.

---

> > ### Author Response · Authors · 2025-08-06
> > **Dear Reviewer YWXi**
> >
> > Dear Reviewer YWXi,
> >
> > We hope our rebuttal has addressed your concerns and questions. If any points remain unclear or require further discussion, we would be grateful if you could let us know. Thank you very much for your time and consideration.
> >
> > Best regards,
> >
> > The Authors

---

> > ### Comment · Reviewer_YWXi · 2025-08-06
> >
> > Thank you for the detailed response.
> >
> > I am still not convinced that the exponential convergence result for regularized MFG is not an expected result. As in regularized MFG, you still have some monotonicity like condition, so I still think the proof can be straightforwardly extended from prior works like Cen et al.
> >
> > Although Cen et al used an extra gradient version of the mirror descent, but it can be seen as an approximation of the proximal algorithm. So it’s not very different from your algorithm, beside specific choice of the regularizers

---

> > > ### Author Response · Authors · 2025-08-07
> > >
> > > Thank you for your continued engagement. We respectfully disagree that our exponential‐rate proof is a straightforward extension of Cen et al.'s work, for two fundamental reasons:
> > >
> > > > `I am still not convinced that the exponential convergence result for regularized MFG is not an expected result. As in regularized MFG, you still have some monotonicity-like condition, so I still think the proof can be straightforwardly extended from prior works like Cen et al.`
> > >
> > > First, Cen et al. study a **zero-sum Markov game**, whereas we address a **Mean-Field Game**.  In zero-sum games, each player's policy is independent and can be optimized individually.  In an MFG, by contrast, the (infinitely many) agents share a population distribution $\mu = m\[\pi]$ that is passively determined by the representative player's policy $\pi$ and state.  This coupling introduces a fixed-point structure　(see Def. 2.2 (ii)) and joint dynamics between policy and population distribution, which does not arise in zero-sum Markov games and presents significant new analytical challenges.
> > >
> > > Second, the **monotonicity** assumptions differ qualitatively.  In Cen et al., monotonicity reduces to **zero-sum structure** (a special case of monotonicity).  In our MFG setting, we impose Assumption 2.3, a **weak monotonicity** —which is itself nonlinear in $\pi$ because $\mu=m\[\pi]$ depends on $\pi$. This nonlinearity and coupling complicate both the algorithmic updates and the convergence analysis.
> > >
> > > While your comment that `it's not very different from your algorithm` is not incorrect at a high level, the convergence proof must be substantially revised to account for these differences.  As noted in **Remark 4.6**, in an MFG, the Q-function $Q\_h^{\lambda,\sigma}(s, a,\pi,\mu)$ depends on every stage's policy $\pi_h$. In contrast, in a Markov game, it only depends on future policies via dynamic programming.  We must therefore develop new contraction arguments and carefully control the $\mu$-dependence of the Bellman updates—steps that go well beyond a "straightforward" mirror‐descent extension of Cen et al.
> > >
> > > For these reasons, we believe our proof is not a trivial extension, and the techniques developed are necessary to handle the unique challenges posed by the MFG setting.

---

### Official Review · Reviewer_K4bh · 2025-07-02

**Clarity:** 3
**Significance:** 3
**Originality:** 3
**Rating:** 5
**Confidence:** 4

**Summary:**

This paper presents a novel algorithmic framework for achieving last-iterate convergence in mean-field games under the assumption of non-strict monotonicity. Current methods often require stricter assumptions like contractivity or strict monotonicity, or only guarantee the convergence of time-averaged policies. The authors' approach is twofold. First, they introduce a Proximal Point method with KL-divergence regularization. They provide a proof that the last iterate of this method asymptotically converges to a mean-field equilibrium, a result established without requiring strict monotonicity. Second, they observe that each update of this PP method is equivalent to finding the equilibrium of a KL-regularized MFG. They then show that this regularized subproblem can be solved by the Regularized Mirror Descent  algorithm with an exponential last-iterate convergence rate, which is a significant improvement over previous polynomial-rate or time-averaged results. Based on these theoretical insights, the paper proposes the Approximate Proximal-Point algorithm, which uses a finite number of RMD steps to approximate the PP update, and demonstrates its empirical convergence on a standard MFG benchmark.

**Questions:**

1.  The APP algorithm's performance seems dependent on $\tau$, the number of inner RMD iterations. The experiments show a difference in performance between $\tau=10$ and $\tau=1000$. What is the primary theoretical obstacle to deriving a convergence guarantee for the combined APP algorithm? How does the approximation error from using a finite $\tau$ affect the stability and convergence of the outer PP loop?
2.  Could the authors elaborate on the technical challenges specific to MFGs that complicate the convergence analysis? The paper correctly points out that the mean-field dependency, where the policy iterate $\pi^k$ influences its own update through the mean-field $\mu^k = m[\pi^k]$, is a key difficulty not present in standard N-player games. Could you provide more intuition on how the use of the Łojasiewicz inequality (for the PP proof) and the detailed value function analysis (for the RMD proof) were tailored to overcome this specific challenge?
3. Could the authors discuss how the RMD differentiates from that in zero-sum game (see e.g. [1] [2] ). Moreover, if the feedbacks are potentially delayed or asynchronous, could the authors give intuition on how to adapt the algorithms non-trivially in these scenarios (see e.g. [3])? A discussion clarifying the relationship to these works would further strengthen the paper's positioning.
3.  How should one select the regularization parameter $\lambda$ and the number of inner iterations $\tau$ in practice? The parameter $\lambda$ controls the gap between the regularized and unregularized equilibria, while $\tau$ controls the accuracy of the PP update. Is there a theoretical relationship between these parameters that would guarantee convergence for the overall APP algorithm?



**Numerical Scores:**

* **Quality & Technical Soundness:** 4
* **Clarity:** 3
* **Originality & Significance:** 3
* **Overall Score:** 4
* **Confidence:** 4





[1] Cen, S., Wei, Y., & Chi, Y. (2021). Fast policy extragradient methods for competitive games with entropy regularization. Advances in Neural Information Processing Systems, 34, 27952-27964.

[2] Park, C., Zhang, K., & Ozdaglar, A. (2023). Multi-player zero-sum markov games with networked separable interactions. Advances in Neural Information Processing Systems, 36, 37354-37369.

[3] Ao, R., Cen, S., & Chi, Y. (2023, January). Asynchronous Gradient Play in Zero-Sum Multi-agent Games. In International Conference on Learning Representations (ICLR).

**Ethical Concerns:**

["NO or VERY MINOR ethics concerns only"]

**Final Justification:**

After carefully reviewing the rebuttal and author responses, I am raising my score from 4 to 5.
Resolved Issues:

The authors provided clear explanations for the theoretical obstacles in proving APP convergence, particularly the challenge of vanishing regularization mass as $\eta\to 0$
They clarified the key distinctions between MFG and zero-sum game settings, especially regarding asymmetric vs. symmetric regularization
They demonstrated understanding of how their techniques (Łojasiewicz inequality, value function analysis) address the unique challenges in MFGs
They committed to adding discussions on connections to related work in multi-agent learning

Remaining Issues:

Lack of formal convergence guarantee for the practical APP algorithm remains the primary limitation
The model-based setting assumption still limits direct applicability to model-free RL scenarios

Weight Assessment:
The theoretical contributions (PP convergence under non-strict monotonicity, exponential RMD convergence) are significant advances that outweigh the remaining limitations. The authors acknowledge these limitations honestly and provide a clear research direction. The promised additions to contextualize their work within the broader literature will strengthen the paper's impact.
Recommendation: Accept. This paper makes solid theoretical contributions to an important problem and presents a well-motivated algorithm, despite some remaining theoretical gaps.

**Limitations:**

The authors provide a good discussion of the paper's limitations in Section 6, correctly identifying the model-based setting, the focus on iteration complexity over per-iteration computational cost, and the lack of analysis with function approximation. The most significant limitation, which is mentioned as a future goal but not framed as a limitation, is the absence of a convergence proof for the main APP algorithm. Moreover, it would be more concrete to discuss the relationship between the RMD in the mean-field game with that in Zero-sum Games and its adaptivity to the asynchronous multi-player scenarios, as mentioned before. A more direct discussion of the challenges involved in bridging this theoretical gap would strengthen the paper.

**Quality:**

4

**Strengths And Weaknesses:**

**Strengths:**

1.  **Addresses the Problem Under Weaker Assumptions:** The paper makes a contribution by tackling last-iterate convergence under non-strict monotonicity. This is a more general and practical assumption than the strict monotonicity or contractivity required by much of the prior work, and it correctly includes important cases, such as games with symmetric transitions, where strict monotonicity fails.

2.  **Theoretical Improvements:** The paper delivers two important theoretical results. The first is the asymptotic last-iterate convergence of the PP method (Theorem 3.1) in this setting. The second, and perhaps more impactful, is the exponential convergence rate established for RMD in solving the regularized MFG subproblems (Theorem 4.3), which improves upon previous results that showed polynomial or time-averaged convergence.

3.  **Well-Motivated and Elegant Algorithm Design:** The proposed APP algorithm is a well-motivated and practical method. The insight to frame the problem as approximating an intractable PP update by using RMD to solve a sequence of regularized MFGs is elegant.

**Weaknesses:**

1.  **Lack of Convergence Guarantee for the Proposed APP Algorithm:** While the paper provides convergence guarantees for the idealized PP method (asymptotic) and the RMD subproblem solver (exponential), it does not provide a formal convergence guarantee or rate for the final, practical APP algorithm (Algorithm 1). APP approximates the PP update with a finite number of RMD steps, introducing an error that is not theoretically analyzed. The authors state that deriving a convergence rate for APP is a conjecture and a future task.

2.  **Reliance on a Model-Based Setting:** The entire analysis is conducted in a model-based setting, where the transition kernels and reward functions are assumed to be known. This assumption limits the direct applicability of the theoretical results to many reinforcement learning scenarios where the agent must learn from data without a model of the environment.

---

> ### Author Rebuttal · Authors · 2025-07-30
>
> Thank you for reviewing our work. We respond to your questions below.
>
> > What is the primary theoretical obstacle to deriving a convergence guarantee for the combined APP algorithm?
>
> In our current analysis of the inner RMD iterations, the allowable step size $\eta^\ast$ must shrink towards zero as $\sigma_{\textup{min}}$ becomes small. Indeed, an equilibrium of the MFG $\pi^\star$ may assign zero probability to some actions, so $\pi^\star_{\textup{min}}=0$. Hence, any convergent APP would drive some $\sigma^k_{\textup{min}}\to 0$, leaving no positive $\eta^\ast$ that satisfies our bounds. Overcoming this requires new techniques to handle vanishing regularization mass.
>
> > How does the approximation error from using a finite $\tau$ affect the stability and convergence of the outer PP loop?
>
> A precise theoretical bound is challenging, but the outer loop remains stable in experiments even for $\tau\approx5$. However, when the contraction factor $(1-\lambda\eta)$ is close to 1, a small $\tau$ can lead to instability in the outer PP loop. We will note this trade-off in the revision.
>
> > Could you provide more intuition on how the use of the Łojasiewicz inequality (for the PP proof) and the detailed value function analysis (for the RMD proof) were tailored to overcome this specific challenge?
>
> Mean-field games are challenging because, even if the policy $\pi$ acts optimally, the mean field $\mu$ merely follows passively and does not itself move optimally towards $\mu^\star$. Our key innovation is to track only the policy updates: the Łojasiewicz inequality quantifies how changes in only $\pi^t$ affect the cumulative reward $J$, allowing us to avoid explicit tracking of $\mu^t$.
>
> > Could the authors discuss how the RMD differentiates from that in zero-sum game (see e.g. [1] [2]).
>
> In zero-sum games, both players’ policies receive explicit regularization terms. By contrast, in an MFG, only the single-agent policy $\pi$ is directly regularized, and the mean field $\mu$ does not appear under the KL term. This asymmetry leads to a single-player fixed-point rather than a saddle-point structure.
>
> > Moreover, if the feedbacks are potentially delayed or asynchronous, could the authors give intuition on how to adapt the algorithms non-trivially in these scenarios (see e.g. [3])?
>
> Conceptually, one can adapt RMD to delayed feedback by evaluating the Q-function $Q_h^{\lambda,\sigma}(s,a,\pi^t,\mu^t)$ at a delayed index $\kappa^{(t)}$, i.e. $Q_h^{\lambda,\sigma}(s,a,\pi^{\kappa^{(t)}},\mu^{\kappa^{(t)}})$. For asynchronous updates one might let $\kappa^{(t)}$ depend on the current $\mu$, but the precise feedback model would need to be specified. We will add a brief discussion of this in the revision.
>
> > How should one select the regularization parameter $\lambda$ and the number of inner iterations $\tau$ in practice? The parameter $\lambda$ controls the gap between the regularized and unregularized equilibria, while $\tau$ controls the accuracy of the PP update.
>
> In practice, one should choose $\lambda<1$ as large as possible—so that the inner loop remains small—while reducing $\eta$ to maintain convergence as to satisfy (D.5). This choice allows a small number of inner iterations $\tau$ while keeping the outer loop stable.
>
> > Is there a theoretical relationship between these parameters that would guarantee convergence for the overall APP algorithm?
>
> Due to the abovementioned obstacle, we have no theoretical guarantee linking $\lambda$, $\tau$, and $\eta$ for the complete APP algorithm.  Techniques in monotone games [Theorem 5.8, Abe et al. 2024] might yield such parameter relationships; we leave this to future work.

---

> > ### Comment · Reviewer_K4bh · 2025-08-04
> >
> > Thank you for the thorough clarification of the theoretical obstacles in deriving convergence guarantees for APP and the insightful explanation of how the Łojasiewicz inequality handles the mean-field dependency. The distinctions between MFG's asymmetric regularization and zero-sum games' symmetric structure are now much clearer. I appreciate these thoughtful responses and look forward to the enhanced discussion in the revision, which will better position this work within the broader multi-agent learning literature.

---

> ### Author Response · Authors · 2025-08-04
>
> Thank you very much for your comments. We are happy to hear that our rebuttal helped clarify some points about our work.
> In the revised manuscript, we will expand the discussion to clearly distinguish the asymmetric regularization in mean-field games from the symmetric structure of zero-sum games.

---

> > ### Comment · Reviewer_K4bh · 2025-08-04
> >
> > Thank you for the acknowledgment. I appreciate your willingness to expand the theoretical discussions. The conceptual adaptation to delayed feedback settings you outlined (evaluating Q at randomly delayed index k-d) was particularly intriguing - such connections could open interesting avenues for future work in this area.

---

### Official Review · Reviewer_Nj5a · 2025-07-03

**Clarity:** 3
**Significance:** 2
**Originality:** 2
**Rating:** 4
**Confidence:** 4

**Summary:**

The paper aims to understand convergence guarantees for monotone finite-horizon MFGs. Existing results have some shortcomings, such as suboptimal rates, assumptions of strict monotonicity, not providing last-iterate convergence, and requiring regularization. This work proves a linear convergence bound for monotone MFGs for regularized MD, improving upon past work. This result is suggested to imply convergence to non-regularized NE by showing that it asmpyotically approximates an implicit proximal point update rule. Experimental evaluations are provided.

**Questions:**

- Could assumption 2.3 also be generalized to the case where the reward function $r$ depends on the mean-field flow over states and actions? That is, when $\mu_h^\pi(s,a)$ is the population flow on states and actions induced by policy $\pi$. In some MFGs, the population's actions distribution is also significant.

- Theorem 4.3 and correspondingly the update rule (4.4) tackle a similar setting as in Zhang et al., 2024. However, while their analysis only yields sublinear convergence,

- The continuous time analysis of Perolat et al., 2021 is more straightforward than the analysis of the work, although it does not yield the more important discrete time convergence. I am curious to what extent the discretization of the ODE (corresponding to the (O)MD update) proposed by Perolate et al. can be used to still obtain a discrete time result. Clearly, in this work, the learning rate $\eta$ is quite small, so the results might already be in the regime where an ODE is discretized. This is already somewhat suggested by Theorem 4.4, however, could the analysis directly incorporate the ODE up to a discretization error? To what extent does the analysis differ from standard ODE discretization methods that are used in optimisation literature?

- The paper comes quite close to deriving a rate for the unregularized problem via APP and the RMD convergence. What is the main bottleneck that prevents merging the results to obtain a novel convergence result?

- What are the major bottlenecks when generalizing the results to graphon MFGs?

**Ethical Concerns:**

["NO or VERY MINOR ethics concerns only"]

**Final Justification:**

My questions regarding the paper's results were answered. I appreciate the efforts of the reviewers to add further explanations to their paper. I still have minor concerns regarding the magnitude of constants, and the authors have promised to expand on this.

Overall, I keep my positive score.

**Limitations:**

Discussed in the paper.

**Paper Formatting Concerns:**

No major concerns, for minor notes see above.

**Quality:**

3

**Strengths And Weaknesses:**

## Strengths

I think the analysis of the work is interesting and potentially a good contribution to the understanding of MFGs and MF reinforcement learning. It is a clean result that is also meaningful for many practical problems.

The paper is well-written, and the theoretical results are easy to follow. The commentary on the theoretical developments compared to existing work on MFGs was helpful.

## Weaknesses:

I list some concerns below, I am happy to discuss further following the responses of the authors.

From the theory side, one major concern is the convergence rate that has been proved. The exponential convergence of Theorem 4.3 matches intuitively the ODE rate, though in discrete time the fraction is $\left(1 - \frac{\lambda\eta}{2}\right)$, where $\eta < \eta^{\*}$. Learning rate $\eta^\*$ defined in D.5 determines the exponential convergence rate, and $\eta^\*$ can be very small. By tracing the results and constants, it seems $\eta^\*$ will scale with $e^{-H\lambda^{-1}}$ and $(-\log \sigma_{min})^{-1}$, which might be extremely small. In this regard, while the convergence result is indeed linear, it is difficult to compare it with the sublinear rate of Zhang et al. 2023. Furthermore, since the convergence of APP is not known, one would typically set $\lambda$ to be small, which exponentially worsens the linear rate.

Related to this, at some point in the main text, mentioning the dependence of $\eta^*$ on the problem parameters might be helpful, at least asymptotically. It is currently somewhat difficult to understand how the rate of linear convergence changes depending on the problem parameters such as $|\mathcal{S}|, |\mathcal{A}|, H, \sigma_{min}$.

Regarding the presentation of the results, while I like the exposition, it was not clear if the PP method had implications beyond being a theoretically illuminating tool. As far as I understand, in the current state of the work the convergence of APP is not known, although MD seems to asymptotically approximate PP.

Finally, I am not fully convinced regarding the strict monotonicity assumption. Looking at the work due to Zhang et al. 2023, it seems that the strict monotonicity assumption is not required for the convergence analysis, but rather for the uniqueness of the regularised equilibrium. Does Theorem 4.3 not also implicitly assume that there exists a unique equilibrium given by $\bar{\omega}^{\*}$?

Minor points:

- potential formatting issue at Figure 1
- Difficult to read axis labels in Figure 2-a

---

> ### Author Rebuttal · Authors · 2025-07-30
>
> Thank you for your careful review and your interest in convergence guarantees for MFGs. We are happy to address your comments below. First, we will add a reply to your comment in the Weakness section.
> > it is difficult to compare it with the sublinear rate of Zhang et al. 2023.
>
> First, Zhang et al. only show convergence of averages over iterations of policy $\pi^t$, whereas we prove convergence of $\pi^t$ itself (last-iterate convergence; LIC). RMDs exhibit exponentially LIC, and our results align with this phenomenon.
>
>
> Let us compare our LIC results with the averaged convergence results of Zhang et al., at least in the asymptotic regime of sufficiently large $t$. Our exponential rate is tighter than the sublinear bound of Zhang et al. 2023. In contrast, as you know, Zhang et al.’s bound for finite $t$ may be numerically smaller since the constant inside our exponent could be pretty small. We will add a remark on this trade‑off in the revised manuscript.
> > Looking at the work due to Zhang et al. 2023, it seems that the strict monotonicity assumption is not required for the convergence analysis, but rather for the uniqueness of the regularised equilibrium. Does Theorem 4.3 not also implicitly assume that there exists a unique equilibrium given by $\bar{\omega}^*$?
>
>  Your observation is correct. Our key technical step to remove the strictness of monotonicity is that non‑strict monotonicity guarantees the uniqueness of the KL‑regularized equilibrium. Therefore, in Theorem 4.3, uniqueness is not an extra assumption but a derived conclusion.
> Compared to Zhang et al. (2023), we emphasize that Zhang’s convergence analysis is carried out **only** under the entropy **regularization** on the MFG—a far stronger condition than non‑strict monotonicity alone. They do not obtain any last‑iterate convergence guarantee in the absence of regularization. By contrast, our Theorem 3.1 establishes last‑iterate convergence for the **unregularized** MFG under only non‑strict monotonicity.
>
> We will next give you an answer to your questions.
> > Could assumption 2.3 also be generalized to the case where the reward function $r$ depends on the mean‑field flow over states and actions? That is, when $\mu_h^\pi(s,a)$ is the population flow on states and actions induced by policy $\pi$. In some MFGs, the population’s actions distribution is also significant.
>
>  Conceptually, such a generalization appears feasible but depends on the exact formulation. We consider that your suggestion of replacing $\mu_h(s)$ by $\mu_h(s, a)$ requires redefining the population flow in equation (2.1), which effectively changes the entire MFG setup. We believe that our convergence arguments would extend without significant changes with an MDP-style redefinition, but we have not yet formalized this. We will include a discussion of this extension in the revision.
> > Could the analysis directly incorporate the ODE up to a discretization error? To what extent does the analysis differ from standard ODE discretization methods used in optimization literature?
>
>  Yes, we could go up through equation (4.6). We follow an ODE‑style argument (akin to Theorem 4.4). The novel part of our proof of Theorem 4.3 lies in bounding the discretization error (see the “blue box” in (4.6)).
> > The paper comes quite close to deriving a rate for the unregularized problem via APP and the RMD convergence. What is the main bottleneck that prevents merging the results to obtain a novel convergence result?
>
>  As you note, the main bottleneck is that $\sigma_{\textup{min}}$ may tend to zero, which forces the allowable step size $\eta^\ast$ to zero. Indeed, an MFG equilibrium $\pi^\ast$ can assign zero probability to some actions, so $\pi^\ast_{\textup{min}}(a\mid s)=0$. Hence, if APP converges, some $\sigma^k(s)$ would converge to zero, and no positive $\eta^\ast$ would satisfy our current analysis. Overcoming this requires new ideas to handle vanishing mass.
> > What are the major bottlenecks when generalizing the results to graphon MFGs?
>
>  We do not anticipate significant difficulties here. Our problem setting for the convergence analysis does not conflict with the graphon generalisation as discussed in the literature. Therefore, it can be applied to graphon MFGs without introducing any new technical hurdles.. Accordingly, our results should extend naturally to graphon MFGs, albeit with more cumbersome notation.

---

> > ### Comment · Reviewer_Nj5a · 2025-08-03
> >
> > Thank you for your responses. I appreciate the future modifications to the text, in particular, a discussion of the effect of constants on the convergence rate for finite $t$. I will maintain my (positive) score for now.

---

> > > ### Author Response · Authors · 2025-08-04
> > >
> > > Thank you again for maintaining your positive score; we really appreciate your support.

---

### Author Response · Authors · 2025-08-09
**Global Comments by the author(s)**

We thank all reviewers for their constructive feedback. Below, we summarize how each major theme raised by reviewers has been addressed:

- **Novelty of the Exponential‐Rate Bound** (Reviewer Nj5a)
  Discussion with Nj5a clarified that splitting the analysis into the **finite-$t$ regime** and the **asymptotic large-$t$ regime** makes the novelty of our exponential bound in Theorem 4.3 clear. For a large iteration $t$, our rate strictly improves on previous sublinear bounds; for finite $t$, we will add remarks on the trade-off introduced by the exponent's constants.

- **Proof Challenges from MFG vs Zero-Sum (Markov) Games** (Reviewers K4bh & YWXi)
  Both K4bh and YWXi observed formal similarities to mirror-descent analyses in zero-sum games, but as highlighted in Remark 4.6, the **population coupling** $\mu=m[\pi]$ and our **non-strict monotonicity** (Assumption 2.3) introduce a nonlinear fixed-point structure not present in standard Markov games. We have detailed how Łojasiewicz's inequality and new contraction arguments control the $\mu$-dependence of the Bellman updates—steps that go well beyond a "straightforward" extension of Cen et al.

- **Comparison to Other MFG Optimization Methods** (Reviewer GbwY)
  GbwY urged us to contrast our "global" guarantees with **local** convergence methods (e.g., Guo et al. 2024). We explain that those require an a priori closeness to equilibrium, whereas our results need only **verifiable** conditions (monotonicity and full support) to ensure convergence of both PP and RMD.

- **Connecting $D_{\mu^*}$ to Exploitability** (Reviewer z4KT)
  z4KT's question led us to formalize the bound
  $$\operatorname{Exploit}(\pi^t)=\mathcal{O} \left(\sqrt{D\_{\mu^\ast}(\varpi^\ast,\pi^t)}\right).$$
This makes explicit how our exponential decay in $D_{\mu^*}$ yields exponential exploitability decay.

- **Minor Corrections**
  - We will correct the typo in Line 71 of the Contributions box (it should read *Theorem 3.1* and *Theorem 4.3*).

We believe these revisions and clarifications strengthen the paper's presentation and underscore its contributions.

---

### Decision · Program_Chairs · 2025-09-17

**Decision:**

Accept (poster)

**Comment:**

This work introduces I once looked at Lastry–Lions monotonicity into the MFG setting, and defines a new proximal point-type method to solve it. This work enhances the field of mean field games by introducing a new class of solvable problems. While the practical utility of this condition is somewhat limited, experimentally the result of introducing the proposed algorithm yields to substantial gains in convergence rate and a novel analysis of the last-iterate which justifies acceptance.